# Trajectory Graph Learning: Aligning with Long Trajectories in Reinforcement Learning Without Reward Design

**Yunfan Li**[*]
University of California, Los Angeles
yunfanli@g.ucla.edu

**Eric Liu**
University of Southern California
eliu4913@usc.edu

**Lin F. Yang** [*]
University of California, Los Angeles
linyang@ee.ucla.edu

## Abstract

Reinforcement learning (RL) often relies on manually designed reward functions, which are difficult to specify and can lead to issues such as reward hacking and suboptimal behavior. Alternatives like inverse RL and preference-based RL attempt to infer surrogate rewards from demonstrations or preferences but suffer from ambiguity and distribution mismatch. A more direct approach, inspired by imitation learning, avoids reward modeling by leveraging expert demonstrations. However, most existing methods align actions only at individual states, failing to capture the coherence of long-horizon trajectories.

In this work, we study the problem of directly aligning policies with expert-labeled trajectories to preserve long-horizon behavior without relying on reward signals. Specifically, we aim to learn a policy that maximizes the probability of generating the expert trajectories. Nevertheless, we prove that, in its general form, this trajectory alignment problem is NP-complete. To address this, we propose *Trajectory Graph Learning* (TGL), a framework that leverages structural assumptions commonly satisfied in practice—such as bounded realizability of expert trajectories or a tree-structured MDP. These enable a graph-based policy planning algorithm that computes optimal policies in polynomial time under known dynamics. For settings with unknown dynamics, we develop a sample-efficient algorithm based on UCB-style exploration and establish sub-linear regret. Experiments on grid-world tasks demonstrate that TGL substantially outperforms standard imitation learning methods for long-trajectory planning.

## 1 Introduction

Reinforcement learning (RL) has emerged as a powerful tool with numerous successful applications, ranging from early advancements in robotics and control [Lee et al., 2020] to more recent breakthroughs in self-driving technology [Dosovitskiy et al., 2017] and the fine-tuning of Large Language Models (LLMs) [Ouyang et al., 2022]. However, the success of RL in real-world applications often relies heavily on the design of the reward function, which typically requires significant prior knowledge. We face challenges such as reward hacking

---

[*]Corresponding authors

39th Conference on Neural Information Processing Systems (NeurIPS 2025).

[Amodei et al., 2016] where unintended behaviors maximize the given reward and reward shaping [Ng et al., 1999] where improperly crafted rewards lead to suboptimal learning.

Since designing scalar, numeric rewards is often impractical for complex real-world tasks, some alternative paradigms have been proposed. Inverse Reinforcement Learning (IRL) [Ng et al., 2000] aims to recover a reward function that explains and justifies an expert's demonstrated behavior—so the agent can reproduce comparable policies without ever being given explicit rewards. More recently, Preference-based Reinforcement Learning (PbRL) [Akrour et al., 2011b, Christiano et al., 2017, Xu et al., 2020, Abdelkareem et al., 2022] replaces hand-crafted numeric rewards with human preferences—typically pairwise or ordinal feedback over trajectories or behaviors—and learns either a surrogate reward or a policy that aligns with those preferences to guide decision-making. However, learning a surrogate reward from demonstrations or preferences does not guarantee alignment with expert intent; ambiguity [Waugh et al., 2013, Lambert and Calandra, 2023, Hu et al., 2023], overfitting [Brown et al., 2019a, Szot et al., 2023] and distribution shift [Fu et al., 2017] all conspire to degrade final policy quality. These empirical and theoretical findings motivate approaches—such as direct trajectory alignment that bypass reward modelling altogether.

Imitation learning [Hussein et al., 2017] circumvents the need for handcrafted reward functions by training policies to replicate expert behavior, typically by learning a mapping from observed states to corresponding expert actions. However, its classical instantiation—behavior cloning (BC) [Torabi et al., 2018]—focuses solely on *state-action pair* alignment, capturing the expert action conditioned on individual states while neglecting the broader trajectory-level structure. As a result, discrepancies between the trajectories generated by the learned policy and the expert demonstrations can accumulate over time, ultimately undermining the preservation of coherent long-horizon behaviors [Ross et al., 2011, Chang et al., 2021]. We illustrate this limitation with a concrete example in Appendix A.2, where BC fails to reliably reproduce entire expert trajectories with probability 1 even in a deterministic environment. This shortcoming motivates our study of *direct trajectory alignment*, which seeks to directly maximize the likelihood of reproducing complete expert trajectories. Such alignment is particularly critical in applications like large language model (LLM) response generation [Zeng et al., 2024] and autonomous driving [Huang et al., 2024], where even small local deviations can propagate into significant degradations in overall quality or safety. To the best of our knowledge, the theoretical foundations of directly aligning policies with expert-labeled trajectories—without relying on reward modeling—remain largely unexplored.

In this paper, we systematically investigate the problem of direct trajectory alignment in reinforcement learning and develop a theoretical framework that enables efficient policy learning through *whole-trajectory* alignment. Our approach eliminates the need for explicit reward modeling and leverages structural assumptions such as reachable expert trajectories in terms of probability and tree-structured MDP to ensure computational tractability. We summarize our key contributions below.

**Our Contributions**

- **Hardness result for direct trajectory alignment.** We prove that the general problem of finding an optimal policy via direct trajectory alignment is NP-complete, by presenting a novel reduction from a classical NP-complete problem. This result highlights the fundamental computational challenge of directly aligning policies with expert trajectories.

- **Theoretical framework: *Trajectory Graph Learning* (TGL).** We introduce *Trajectory Graph Learning* (TGL), a theoretical framework that casts the direct trajectory alignment problem as a maximum weight independent set problem over a trajectory-induced graph. Under structural assumptions—such as bounded realizability of expert trajectories or a tree-structured MDP—we show that the optimal policy can be computed in polynomial time. In the setting with unknown dynamics, we integrate an *upper-confidence bound* (UCB) exploration strategy and design a learning algorithm with provably sub-linear cumulative regret.

- **Empirical validation.** We empirically evaluate TGL on grid-world benchmarks. Our results show that TGL consistently aligns with expert-labeled trajectories more faithfully than standard behavior cloning across various trajectory sets.

**Related work**   We review several classical RL approaches. The first line of work is inverse reinforcement learning (IRL), which infers surrogate rewards from expert demonstrations; the second is preference-based RL (PbRL), which learns from preferences over trajectories; and the third is imitation learning, which directly maps expert demonstrations to policies without reward modeling.

**Inverse RL**   Early work framed IRL as reward reconstruction: Ng et al. [2000]'s linear program formulation and Abbeel and Ng [2004]'s feature expectation matching seek a reward under which the expert is optimal, a paradigm later disambiguated with maximum-entropy regularization [Ziebart et al., 2008]. Deep variants such as Guided Cost Learning [Finn et al., 2016], GAIL [Ho and Ermon, 2016], and AIRL [Fu et al., 2018] scale this two-stage pipeline, yet their rewards remain trustworthy only near the demonstration distribution, leading to policy mis-alignment after exploration. Recent extrapolation and theory papers such as T-REX [Brown et al., 2019b], the finite-sample analysis [Komanduru and Honorio, 2019], and empirical audits of RLHF reward models [Kaplan and et al., 2023] confirm that surrogate rewards do not guarantee alignment, motivating the need to analyze direct trajectory alignment as an alternative.

**PbRL**   From the first simulator-free preference-based policy learning algorithms [Akrour et al., 2011a, Wilson et al., 2012, Busa-Fekete et al., 2013] to the widely cited deep RL from human preferences framework [Christiano et al., 2017], most PbRL methods fit a surrogate reward to pairwise or $K$-wise feedback and then optimize it with standard RL. Recent PbRL algorithms still fit *implicit* reward surrogates whose quality hinges on pre-chosen feature embeddings and Bradley–Terry–Luce (BTL) style preference models. For instance, Saha et al. [2023] established finite-time regret bounds only after projecting trajectories into a hand-crafted feature space, so alignment requires that this embedding fully captures task-relevant differences. The finite-sample analysis of Xu et al. [2020] likewise assumes an unobserved latent reward and proves guarantees under the oracle condition that pairwise labels reflect that hidden function. Empirical pipelines such as Direct Preference Optimisation (DPO) [Rafailov et al., 2023] and Direct PB-PO [An et al., 2023] optimize the same BTL-based surrogate, while more recent theory—SeqRank's principled comparison loss [Zhu et al., 2023], the reward-agnostic RAPT optimiser [Zhan et al., 2023], and best-policy identification from preferences [Agnihotri et al., 2025]—all rely on accurate trajectory embeddings and preference-likelihood calibration to recover near-optimal policies. These dependencies indicate that PbRL, which relies on feature embeddings and preference models, may fail to guarantee alignment with expert behavior—highlighting the need to study the problem of direct trajectory alignment.

**Imitation Learning**   Imitation learning (IL) dispenses with reward design by training a policy to copy expert demonstrations [Hussein et al., 2017]. The dominant variant, behaviour cloning, maps each observed state to the expert's action [Torabi et al., 2018]; because it matches actions *state-by-state*, it incurs covariate shift, so small errors push the learner into unseen states and the mismatch compounds along a trajectory [Ross et al., 2011]. Recent analyses confirm that even with offline fixes such as MILO trajectory-level fidelity remains loosely bounded [Chang et al., 2021]. Thus, no existing IL framework directly optimizes expected alignment with a set of expert-labelled trajectories.

## 2   Problem Formulation

In this section, we first introduce and define the direct trajectory alignment problem. We consider a finite-horizon Markov Decision Process (MDP) $M = (\mathcal{S}, \mathcal{A}, \mathbb{P}, \mathcal{T}_{\text{sel}}, H, \mu)$, where $\mathcal{S}$ is the state space, $\mathcal{A}$ is a finite action space, $\mathbb{P} : \mathcal{S} \times \mathcal{A} \to \Delta(\mathcal{S})$ is the transition kernel, $H$ is the planning horizon (i.e., episode length), and $\mu$ is the initial state distribution. The agent interacts with the environment episodically. In each episode of length $H$, the agent follows a policy $\pi = \{\pi_h\}_{h=1}^{H}$, where each $\pi_h : \mathcal{S} \to \mathcal{A}$ maps a state to an action at time step $h \in [H]$. A policy $\pi$ induces a trajectory $\tau = (s_1, a_1, s_2, a_2, \ldots, s_H, a_H)$, where $s_1 \sim \mu$, $a_1 = \pi_1(s_1)$, $s_2 \sim \mathbb{P}(\cdot \mid s_1, a_1)$, $a_2 = \pi_2(s_2)$, and so on. Let $\mathcal{T}$ denote the set of all possible $H$-length trajectories in the environment. Instead of having a reward function as in traditional RL, we

are given a *Expert-Labeled Trajectory Set*, defined as a small subset $\mathcal{T}_{\text{sel}} = \{\tau_i\}_{i=1}^M \subset \mathcal{T}$ [2], where each $\tau_i$ is a trajectory. Typically, the size of the expert-labeled trajectory set is small relative to the entire trajectory space, i.e., $M \ll |\mathcal{T}|$.

Each policy $\pi$ induces a distribution $d^\pi$ over the space of $H$-length trajectories $\mathcal{T}$, where $d^\pi(\tau) = \mu(s_1) \prod_{h=1}^H \pi(a_h \mid s_h) \cdot \mathbb{P}(s_{h+1} \mid s_h, a_h)$. The objective is to find a policy that maximizes the visitation probability of the trajectories in the expert-labeled set.

$$\pi^* = \arg\max_{\pi \in \Pi} \sum_{\tau \in \mathcal{T}_{\text{sel}}} d^\pi(\tau)$$

Direct trajectory alignment replaces the cumulative reward used in standard RL with an expert-labeled trajectory set as the criterion for policy evaluation. In standard RL, the effectiveness of a learning algorithm is often measured by *regret*, defined as the difference between the cumulative reward of the optimal policy and that of the agent's learned policy over time. Analogously, in our direct trajectory alignment setting, we can define a notion of regret that captures the performance gap in aligning with the expert-labeled trajectories. Specifically, for each round $k \in [K]$, suppose the agent starts from the same initial distribution $\mu$ and executes policy $\pi^k$ to generate a trajectory. The cumulative regret after $K$ rounds is then defined as

$$\text{Regret}(K) = \sum_{k=1}^K \left( \max_{\pi \in \Pi} \sum_{\tau \in \mathcal{T}_{\text{sel}}} d^\pi(\tau) - \sum_{\tau \in \mathcal{T}_{\text{sel}}} d^{\pi_k}(\tau) \right), \tag{1}$$

where $d^\pi(\tau)$ denotes the probability of generating trajectory $\tau$ under policy $\pi$.

**Notation** We use $[n]$ to represent index set $\{1, \cdots n\}$. For $x \in \mathbb{R}$, $\lfloor x \rfloor$ represents the largest integer not exceeding $x$ and $\lceil x \rceil$ represents the smallest integer exceeding $x$. We use $O$ to represent leading orders in asymptotic upper bounds and $\widetilde{O}$ to hide the polylog factors. For a finite set $\mathcal{A}$, we denote the cardinality of $\mathcal{A}$ by $|\mathcal{A}|$.

## 3 Hardness of General Direct Trajectory Alignment Problem

In this section, we will show the hardness of solving a general direct trajectory alignment problem. We will show that getting the optimal policy in a general direct trajectory alignment problem is equivalent to solving a **Maximum-Weight Independent Set (MWIS)** problem in a graph. First, we will introduce some concepts here.

**Definition 1** (Conflict in State-Action Pair). *Two state-action pairs $(s, a)$ and $(s', a')$ are in* conflict *if they share the same state but have different actions:*

$$s = s' \quad and \quad a \neq a'.$$

**Definition 2** (Conflict in Trajectories). *Consider two trajectories of length $H$:*

$$\tau_1 = (s_1, a_1, \ldots, s_H, a_H), \quad \tau_2 = (s_1', a_1', \ldots, s_H', a_H').$$

*They are in* conflict *if there exists $h \in \{1, \ldots, H\}$ such that*

$$s_h = s_h' \quad and \quad a_h \neq a_h'.$$

**Remark.** *If $\tau_1$ and $\tau_2$ conflict, and $\tau_1$ and $\tau_3$ do not conflict, it does not imply that $\tau_2$ and $\tau_3$ are conflict-free.*

**Definition 3** (Trajectory-Induced Policy Set). *For a trajectory $\tau = (s_1, a_1, \ldots, s_H, a_H)$, its trajectory-induced policy set $\pi^\tau$ contains all policies $\pi = \{\pi_h\}_{h=1}^H$ satisfying $\pi_h(s_h) = a_h$ for each $h$, while $\pi_h(s')$ is arbitrary for $s' \neq s_h$:*

$$\pi^\tau = \left\{ \pi \mid \pi_h(s_h) = a_h \ \forall h = 1, \ldots, H; \ \pi_h(s') \ arbitrary \ for \ s' \neq s_h \right\}.$$

---

[2] Our setting can be naturally extended to scenarios where the expert-labeled trajectory set is accompanied by scalar feedback, such as scores provided by experts. In this case, we have $\mathcal{T}_{\text{sel}} = \{(\tau_i, y_i)\}_{i=1}^M \subset \mathcal{T} \times \mathcal{Y}$, where each $\tau_i$ denotes a trajectory and $y_i \in \mathcal{Y}$ represents its associated scalar label. Accordingly, the objective should incorporate the feedback by weighting each trajectory with its corresponding label $y_i$.

This set represents all policies that exactly reproduce $\tau$ on its specific states. Since we focus on deterministic policies, conflicts arise when a policy must choose a unique action at the same state, unlike randomized policies which can mix actions.

**Remark.** *If $\tau_1$ and $\tau_2$ conflict, then $\pi^{\tau_1} \cap \pi^{\tau_2} = \emptyset$ because a deterministic policy cannot choose conflicting actions at the same state. Conversely, if a set of trajectories $\{\tau_1, \ldots, \tau_m\}$ are pairwise conflict-free, then $\bigcap_{i=1}^{m} \pi^{\tau_i} \neq \emptyset$.*

For each trajectory $\tau_i = (s_1^{(i)}, a_1^{(i)}, \ldots, s_H^{(i)}, a_H^{(i)})$, let

$$p_i = \mu(s_1^{(i)}) \cdot P_1(s_2^{(i)} \mid s_1^{(i)}, a_1^{(i)}) \cdots P_{H-1}(s_H^{(i)} \mid s_{H-1}^{(i)}, a_{H-1}^{(i)}).$$

If a deterministic policy $\pi$ belongs to $\bigcap_{i=1}^{k} \pi^{\tau_i}$, then

$$P(\text{Agent visits } \{\tau_1, \ldots, \tau_k\} \mid \pi) = \sum_{i=1}^{k} p_i.$$

At the same time we can consider a **Conflict Graph** $G = (V, E, W)$: where the vertex set $V$ represents the trajectories $\tau_1, \tau_2, \cdots, \tau_M$ in $\mathcal{T}_{\text{sel}}$, the edge set $E$ is constructed if any pair of trajectories are conflict. For the weight set $W = \{w_i\}$, we let $w_i = p_i$, the probability product of realizing that trajectory.

Thus, the original problem reduces to the following **Maximum-Weight Independent Set (MWIS)** problem on graph $G$ with weights $p_i$:

$$\max \sum_{i=1}^{k} p_i\, x_i, \quad x_i + x_j \leq 1 \ \ \forall\, (v_i, v_j) \in E, \quad x_i \in \{0, 1\} \ \ \forall i. \tag{2}$$

Let $S = \{v_i : x_i = 1\}$ be the optimal node set; the corresponding policy selects $\pi_h(s_h^{(i)}) = a_h^{(i)}$ for all $v_i \in S$ and $h = 1, \ldots, H$.

It is well known that the **Maximum Weight Independent Set (MWIS)** problem in a graph is NP-complete [Garey and Johnson, 1979]. We have demonstrated that our original problem—finding the optimal policy in the binary-labeled setting—can be reduced to an instance of the **MWIS** problem. This motivates us to explore whether these two problems are equivalent in computational complexity. In fact, we establish the following result:

**Theorem 1.** *The problem of finding the optimal policy in the direct trajectory alignment setting is NP-complete.*

**Remark.** *We prove this by showing reduction from a known NP-complete problem –the **MWIS** problem. We show that any weighted graph can be represented as a subset of trajectories with corresponding probabilities. We achieve this by using a novel BFS-Based trajectory construction to transfer any weighted graph to a subset of trajectories in one MDP. Then we show that good policy implies high weight independent set, which concludes the proof. The details is provided in the Appendix A.3.*

## 4 Trajectory Graph Learning with Known Model

When we face the hardness of finding the exact optimal solution of the general problem in polynomial time implied by Theorem 1, one may think of the path to find approximation solution with polynomial time. Unfortunately, prior work by Johan [1999] shows that the MWIS problem is extremely hard to approximate, establishing that unless P = NP, there is no $\frac{1}{n^{1-\varepsilon}}$-approximation algorithm for MWIS for any fixed $\varepsilon > 0$, where $n$ denotes the number of nodes in the graph. However, we establish the *Trajectory Graph Learning* (TGL) framework with positive results when certain assumptions is added on the MDP or the expert-labeled trajectory set.

**Case 1: Bounded realizability of expert trajectory set**    In the first case, we assume that all trajectories in $\mathcal{T}_{\text{sel}}$ are reachable, meaning they have non-negligible probability under the environment. This is a standard and practical assumption, as trajectories with vanishingly small probability are often ignored or excluded during data collection or preprocessing. As a result, the selected set $\mathcal{T}_{\text{sel}}$ contains only trajectories that the agent has a reasonable chance of encountering.

---
**Algorithm 1** TGL-CP
---
**Require:** Finite-horizon MDP $(\mathcal{S}, \mathcal{A}, P, H)$; Expert-Labeled Trajectory Set $\mathcal{T}_{\text{sel}} = \{\tau_1, \ldots, \tau_M\}$; **MWIS oracle** $\text{MWIS}(G, w) \to S$
**Ensure:** Chosen trajectory subset $S$ and derived policy $\pi$
    /* **Conflict graph** */
1: $V \leftarrow \{v_i \mid \tau_i \in \mathcal{T}_{\text{sel}}\}, \ E \leftarrow \varnothing$
2: **for all** $(\tau_i, \tau_j)$ with $i < j$ **do**
3:      **if** $\exists h : s_h^i = s_h^j \wedge a_h^i \neq a_h^j$ **then**
4:         $E \leftarrow E \cup \{(v_i, v_j)\}$
    /* **Weights** */
5: **for all** $\tau_i \in \mathcal{T}_{\text{sel}}$ **do**
6:      $p_i \leftarrow \prod_{h=1}^{H-1} P\big(s_{h+1}^i \mid s_h^i, a_h^i\big)$
    /*MWIS Oracle */
7: $S \leftarrow \text{MWIS}\big(G, (p_1, \ldots, p_M)\big)$
8: **for all** $v_i \in S$ with $\tau_i = (s_1^i, a_1^i, \ldots, s_H^i, a_H^i)$ **do**
9:      **for** $h = 1$ **to** $H$ **do**
10:        $\pi_h(s_h^i) \leftarrow a_h^i$
11: For states not covered by $\bigcup_{v_i \in S} \tau_i$, set $\pi_h$ via a default rule
12: **return** $S, \pi$
---

**Definition 4** ($\varepsilon$ - Realizable)**.** *We say a trajectory $\tau = (s_1, a_1, s_2, a_2, \ldots, s_H, a_H)$ is $\varepsilon$-Realizable if the probability product $P_1(s_2|s_1, a_1) \cdot P_2(s_3|s_2, a_2) \cdots P_{H-1}(s_H|s_{H-1}, a_{H-1}) \geq \varepsilon$.*

**Example.** In a nearly deterministic environment, where transitions are deterministic or have high probability (e.g., $P(s_{h+1} \mid s_h, a_h) \geq 0.9$ for all steps), many trajectories are $\varepsilon$-realizable with relatively large $\varepsilon$ (e.g., $\varepsilon \approx 0.9^{H-1}$). In contrast, in highly stochastic environments, some trajectories may have exponentially small realization probabilities (e.g., $\sim p^{H-1}$ for small $p$), making them effectively unreachable in practice unless $\varepsilon$ is extremely small.

Then we provide a generic framework **TGL-CP** (***T**rajectory **G**raph **L**earning-**C**onflict **P**lanner*), which is also shown in Algorithm 1. Basically, the algorithm will first construct a conflict graph, where the edges can be determined by checking the conflicts in the expert-labeled trajectory set and the weights can be calculated by the probability product of each trajectory. Then, a MWIS oracle is applied to select a subset of trajectories $S$ that forms a solution to the problem. The desired policy is subsequently obtained by following the actions specified in the trajectories contained in $S$, ensuring the policy aligns with the selected subset of trajectories. To implement this step explicitly, we introduce a simple enumeration-based oracle for MWIS, detailed in Algorithm 3 in Appendix A.1. Then we have the following theorem.

**Theorem 2.** *Assume that any trajectory in the expert-labeled trajectory set $\mathcal{T}_{sel}$ is $\varepsilon_0$-Realizable, then after applying Algorithm 1 and 3, it can return the optimal policy $\pi^*$ with time complexity $O(M^2 H + M^{1/\lfloor 1/\epsilon_0 \rfloor})$.*

**Remark.** *The enumeration-based oracle exploits the lower bound $w(v) \geq \varepsilon_0$ implied by the $\varepsilon_0$-realizable assumption to deduce that any feasible independent set can contain at most $K = \lfloor 1/\varepsilon_0 \rfloor$ vertices. Leveraging this, the oracle exhaustively enumerates all subsets of vertices of size at most $K$, checks each for independence, and calculates their total weight. By keeping track of the heaviest feasible subset encountered, the oracle returns the exact maximum-weight independent set in time polynomial in the graph size, assuming $K$ is treated as a constant.*

**Case 2: Tree-Structured MDP** In the second case, instead of focusing on a general MDP, we instead look for some special structured but very useful MDP setting. The one we will display here is a so-called Tree MDP.

**Definition 5** (Tree MDP)**.** *A Tree MDP is defined as follows:*

*1. No subsequent crossover for different states: For any step $h \in [H]$, for any two different states $s_h^{(1)}$ and $s_h^{(2)}$, the subsequent states after these two states should be different, i.e. if*

*the possible visited states after $s_h^{(1)}$ and $s_h^{(2)}$ are $\sigma(s_h^{(1)})$ and $\sigma(s_h^{(1)})$ respectively, then we have $\sigma(s_h^{(1)}) \cap \sigma(s_h^{(1)}) = \emptyset$.*

2. *No subsequent crossover for different actions: For any step $h \in [H]$, each state $s_h$ has possible actions leading to possible successor states with no merges; i.e. if action $a_1$ leads to possible states $\sigma^{(1)}(s_h)$ and a different action $a_2$ leads to $\sigma^{(2)}(s_h)$, then $\sigma^{(1)}(s_h) \cap \sigma^{(2)}(s_h) = \emptyset$.*

3. *No revisits: once the MDP leaves $s_h$, it never returns to it in future steps.*

*Consequently, the transition graph is a* forward-branching *tree. A* trajectory *is any path from the initial state $s_1$ to a layer-$H$ leaf. Two trajectories conflict if at some time $h$ they coincide in the same state but choose different actions, forming an edge in the conflict graph.* Tree MDPs arise naturally in applications where future decisions unfold independently and paths do not merge. In *LLMs*, decoding strategies like beam search or top-$k$ sampling generate diverse continuations from a prompt, forming a forward tree where each branch represents a distinct sequence [AssemblyAI, 2023]. In *self-driving*, planning algorithms simulate future actions (e.g., turn, accelerate) under constraints that avoid revisiting past states, resulting in a branching structure of possible trajectories [Zhao et al., 2025].

Under the Tree MDP structure, we provide a novel algorithm **TGL-PrunedTree**, which is detailed in Algorithm 4 in Appendix A.1. It is a backward trajectory selection and pruning algorithm tailored for solving the policy optimization problem in a finite-horizon Tree MDP. Given a set of $M$ trajectories $\mathcal{T}_{\text{sel}} = \{\tau_i\}_{i=1}^{M}$, each of fixed length $H$, the algorithm first computes a weight for each trajectory based on the product of transition probabilities along its path. It then proceeds in a backward fashion, from the final timestep $H$ to the root, performing aggregation and pruning at each level. For each state $s_h$ at timestep $h$, it aggregates the weights of trajectories sharing the same action $a_h$ and keeps only the action with the highest total weight. This approach prunes conflicting or suboptimal paths and merges consistent ones, yielding a compatible, high-weights subset of the original trajectories.

**Theorem 3.** *Under the Tree MDP assumption (Definition 5), Algorithm 4 computes an optimal policy $\pi^*$ with time complexity $\mathcal{O}(H \cdot M \cdot |\mathcal{A}|)$.*

**Remark.** *The full proof is provided in Appendix A.5. The key insight here is that, under the Tree MDP structure, the optimal policy can be computed efficiently using a linear-time dynamic programming approach—rather than solving an NP-hard problem as in the general graph case. This result paves the way for analyzing more efficient algorithms, both in terms of time and sample complexity, within learning settings that exhibit tree-like structure.*

## 5 Trajectory Graph Learning with Unknown Model

In real-world scenarios, even when expert trajectories or human-labeled datasets are available, the underlying environment dynamics are typically unknown—that is, the transition matrix $\mathbb{P}$ must still be learned. In this section, we investigate methods for jointly learning the environment and identifying a good policy, a setting commonly referred to as *online learning*. Specifically, we introduce a UCB-based exploration algorithm and provide its corresponding regret analysis.

In the online learning setting, we propose **TGL-UCB** (Algorithm 2), which learns an optimal subset of expert-labeled trajectories by combining Monte Carlo sampling with an Upper Confidence Bound (UCB) exploration strategy [Auer et al., 2002] over a conflict graph. Each node in the graph represents a trajectory, and edges connect conflicting pairs that cannot be realized simultaneously. For each trajectory node $v_i \in V$, we track: $T_i$ (the total number of times a matching policy has been played), $N_i$ (the number of times trajectory $v_i$ has been realized), and $\widehat{\mu}_i = N_i/T_i$ (the empirical realization probability). Initially, for each $v_i$, we generate a policy $\pi_i$ by selecting an arbitrary maximal independent set containing $v_i$ and play each once to initialize estimates. In each round, TGL-UCB computes optimistic upper confidence bounds for all trajectories, invokes an MWIS oracle to select an independent set maximizing the total UCB values, executes the corresponding policy, and updates estimates. This iterative process balances exploration and exploitation, enabling TGL-UCB to progressively refine its trajectory selection and align with expert-labeled demonstrations. Before presenting the main theoretical result, we introduce some essential definitions.

---

**Algorithm 2** TGL-UCB

---

**Require:** Expert-Labeled Trajectory Set $\mathcal{T}_{\text{sel}} = \{\tau_1, \ldots, \tau_M\}$; Total number of rounds $n$;
   **MWIS oracle** $\text{MWIS}(G, w) \to S$
**Ensure:** Chosen trajectory subset $S$ and derived policy $\pi$
   **/* Conflict graph */**
1: $V \leftarrow \{v_i \mid \tau_i \in \mathcal{T}_{\text{sel}}\}$, $E \leftarrow \varnothing$
2: **for all** $(\tau_i, \tau_j)$ with $i < j$ **do**
3:    **if** $\exists h \colon s_h^i = s_h^j \wedge a_h^i \neq a_h^j$ **then**
4:       $E \leftarrow E \cup \{(v_i, v_j)\}$
5: **Initialization:**
6:    For each node (trajectory) $v_i \in V$, use variable $T_i$ as the total number of policies
   played that matches trajectory $v_i$, variable $N_i$ as the times that $v_i$ is sampled so far,
   and variable $\widehat{\mu}_i$ as the current estimated empirical probability of realizing trajectory $\tau_i$,
   where $\widehat{\mu}_i = \frac{N_i}{T_i}$.
7:    For each node $v_i \in V$, select an arbitrary maximum independent $S_i \subset V$ such that
   $v_i \in S_i$, and get the corresponding policy $\pi_i$ from $S_i$.
8:    For each $i \in [M]$, play $\pi_i$ once and update variables $T_i$ and $\widehat{\mu}_i$.
9: **for** $t = M+1, M+2, \cdots, n$ **do**
10:    **for** $i \in [M]$ **do**
11:       Set $u_i = \widehat{\mu}_i + \sqrt{\frac{3 \log t}{2 T_i}}$
12:    Compute $S^+ \leftarrow \text{MWIS}(G, \{u_v\}_{v \in V})$
13:    Play the corresponding policy $\pi^+$ from $S^+$ and update all $T_i'$s, $N_i$'s and $\widehat{\mu}_i$'s.

---

For any independent set $S \subset V$, define $p_S = \sum_{v \in S} p_v$ and the optimal value $p^* = \max_{S \text{ indep.}} p_S$. The set of sub-optimal solutions is

$$\mathcal{S}_{\text{sub}} = \{S \subset V \mid p_S < p^*\}.$$

For each node $v \in V$, define the sub-optimal gaps:

$$\Delta_{\min}^v = p^* - \max\{p_S \mid S \in \mathcal{S}_{\text{sub}}, v \in S\}, \quad \Delta_{\max}^v = p^* - \min\{p_S \mid S \in \mathcal{S}_{\text{sub}}, v \in S\}.$$

The global gap bounds are:

$$\Delta_{\min} = \min_{v \in V} \Delta_{\min}^v, \quad \Delta_{\max} = \max_{v \in V} \Delta_{\max}^v.$$

**Theorem 4.** *The expected regret of Algorithm 2 over $K$ rounds is at most $O\left(\frac{M^3 \log K \cdot \Delta_{\max}}{\Delta_{\min}^2}\right)$.*

**Remark.** *This theorem establishes a gap-dependent regret bound and guarantees sub-linear regret, indicating that efficient learning is achievable in our setting given access to a good MWIS oracle. The detailed proof is provided in Appendix A.6. The key idea is to reduce our UCB-based graph learning problem to a classical combinatorial multi-armed bandit (CMAB) problem, enabling the application of standard analysis techniques.*

## 6   Experiments

To further demonstrate the effectiveness of our method, we conduct experiments comparing our TGL-UCB approach with the classical imitation learning baseline, behavior cloning. This experiment is deliberately designed to stress trajectory-level alignment.

**Environment.** All experiments use a $4 \times 4$ FROZEN LAKE environment [Brockman et al., 2016] modified to make *holes* non-terminating and yield a $-1$ reward. The goal square returns $+1$ and ends the episode; otherwise the horizon is $H = 10$. On every step the intended action is replaced by a uniformly–random valid action with probability 0.10. Observations are one-hot state vectors and actions are the four cardinal moves.

**Demonstrations.** For each condition in Table 1, we build a set $\mathcal{D} = \{\tau_i\}$ of "expert" trajectories that serve as offline data: these trajectories are produced by a deterministic expert or by its stochastic variant. By our definition of trajectory conflicts, the trajectories produced by any deterministic agent are not conflicting, which is why we sample some trajectories from a stochastic agent to induce more conflicts. We then tested various

Table 1: Trajectory match probability (↑ better) over 10,000 episodes with 95% CIs (±). Each row shows how many deterministic and stochastic demos are in $\mathcal{D}$.

| Expert-Labeled Set | TGL-UCB (Ours) | BC | PPO Expert |
|---|---|---|---|
| 15 det. & 5 stoch. | **0.794 ± 0.008** | 0.775 ± 0.008 | 0.778 ± 0.008 |
| 10 det. & 10 stoch. | **0.801 ± 0.008** | 0.778 ± 0.008 | 0.800 ± 0.008 |
| 5 det. & 15 stoch. | **0.801 ± 0.008** | 0.771 ± 0.008 | 0.793 ± 0.008 |
| 10 det. & 5 stoch. | **0.814 ± 0.008** | 0.793 ± 0.008 | 0.803 ± 0.008 |
| 8 det. & 7 stoch. | **0.810 ± 0.008** | 0.774 ± 0.008 | 0.801 ± 0.008 |
| 5 det. & 10 stoch. | 0.779 ± 0.008 | 0.758 ± 0.009 | **0.780 ± 0.008** |
| 10 det. only | **0.778 ± 0.008** | 0.753 ± 0.009 | 0.775 ± 0.009 |
| 5 det. & 5 stoch. | **0.740 ± 0.009** | 0.713 ± 0.009 | 0.735 ± 0.009 |
| 10 stoch. only | 0.738 ± 0.009 | 0.714 ± 0.009 | **0.739 ± 0.009** |
| 5 det. only | **0.730 ± 0.009** | 0.703 ± 0.009 | 0.713 ± 0.009 |
| 5 stoch. only | **0.675 ± 0.009** | 0.673 ± 0.009 | 0.661 ± 0.009 |
| 3 det. only | **0.672 ± 0.009** | **0.672 ± 0.009** | 0.670 ± 0.009 |
| 3 stoch. only | **0.657 ± 0.009** | 0.653 ± 0.010 | 0.648 ± 0.010 |
| 1 det. only | **0.634 ± 0.010** | 0.627 ± 0.010 | 0.621 ± 0.010 |

trajectory set sizes (from 1 to 20) and various deterministic/stochastic compositions. No further interaction with the environment is allowed during training.

**Methods compared.**

- **TGL (ours).** Algorithm 2 (TGL-UCB) is run on $\mathcal{D}$ with initial number of samples $m_0 = 10$, and successful probability $\delta = 0.9$. We solve for the exact MWIS solution in the algorithm (since our environment is small). The resulting maximum-weight independent set $\mathcal{S}^+$ is turned into a time-indexed lookup policy as described in Section 5 of the paper.
- **Behavioural Cloning (BC).** We train the supervised learner `imitation.algorithms.bc` from the opens ource imitation library [Gleave et al., 2022], with batch size of 8 for 20 epochs on the same trajectory set.
- **PPO Expert.** The reference expert policy used to generate the deterministic demonstrations (trained for 50 k steps with PPO).

**Metric.** Because numerical rewards are absent during training, we evaluate policies by the probability that a rollout $\hat{\tau}$ *exactly matches* one of the demonstration trajectories:

$$P_{\text{match}}(\pi) = \Pr_{\hat{\tau} \sim \pi} \left[ \hat{\tau} \in \mathcal{D} \right].$$

For each policy we execute 10 000 episodes with fixed seed and report the empirical match frequency. This empirical trajectory match probability metric directly measures the objective of direct trajectory alignment (DTA): replicate the expert trajectories, rather than maximizing a surrogate reward. Each episode can be viewed as a Bernoulli trial that either matches a demonstration trajectory or not, so the empirical match probability is a binomial proportion.

**Error Bars.** We report the 95% confidence intervals for each configuration using Wald intervals computed from the 10,000 trials. In the worst case $p = 0.5$, the standard error is $\sqrt{(0.5(1 - 0.5)/10000} = 0.005$, yielding a ≈ 0.01 wide 95% interval. In practice, we report the error bars based on the empirical $\hat{p}$ from each cell.

**Discussion.** Across every demo composition, TGL-UCB consistently matches or exceeds the behavioural-cloning baseline and often surpasses the PPO expert, despite never observing rewards. The gains are the most pronounced when demonstrations are scarce or highly mixed, highlighting the benefit of directly learning from trajectories.

**Limitations and Future Work.** Our study was done using a single finite and discrete $4 \times 4$ FROZEN LAKE gridworld-like environment. The current theory and algorithmic guarantees target finite MDPs with a finite exppert set and do not extend to continuous MDPs. Scaling grid size increases the number of states but does not qualitatively change the conflict structure, so the results here primarily serve to validate the trajectory-alignment mechanism and TGL-UCB algorithm. TGL-UCB is also more computationally expensive than Behavioral Cloning due to solving MWIS over the conflict graph being the main bottleneck. we therefore capped the number of iterations at 50 (which is sufficient for the $4 \times 4$ FROZEN LAKE environment.

Future work includes improving the computational efficiency of our methods, extending the DTA framework to continuous MDPs, and extending the experiments to evaluate alignment without reward modeling in more complex or richer environments.

Also, we did not evaluate history-conditioned BC or recurrent sequence models because the environment is fully observed. A next step is to compare TGL to history-conditioned BC or recurrent models in controlled partially observed tasks and to develop a history-aware TGL that builds conflict graphs over history features or trajectory segments.

## 7 Conclusions and Future Work

This work introduces *Trajectory Graph Learning* (TGL), a novel framework for trajectory-level policy alignment that bypasses reward modeling. By leveraging structural assumptions often met in practice, TGL enables efficient and theoretically grounded planning in both known and unknown environments. Our theoretical results highlight the inherent complexity of direct trajectory imitation, while our algorithms demonstrate strong empirical gains over conventional imitation learning methods. These findings underscore the promise of structure-aware trajectory alignment for reliable long-horizon decision-making in reinforcement learning.

A key direction going forward is extending the theory to continuous MDP settings and the development of trajectory-level RL benchmarks, specifically designed to evaluate alignment without reward modeling. This may involve adapting existing environments or designing new ones where trajectory sets can be meaningfully constructed, compared, and evaluated. Building such benchmarks is non-trivial and will likely require substantial engineering effort, especially in continuous or high-dimensional domains such as robotics, autonomous driving, or natural language generation. Additionally, improving the efficiency of TGL, through approximate solvers, amortized inference, or scalable graph-based methods, will be crucial for extending its applicability to larger and more complex settings.

## Acknowledgements

YL is supported in part by NSF grant 2221871 and an Amazon AI Fellowship. LY is supported in part by NSF grant 2221871, and an Amazon Faculty Award.

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

## A   Technical Appendices and Supplementary Material

### A.1   Remaining Algorithm pseudocodes

We provide the remaining algorithms in this section.

---

**Algorithm 3** ENUMMWIS($G, w, \epsilon_0$)

---

**Require:** Conflict graph $G = (V, E)$ with $|V| = M$; weights $w : V \rightarrow (0, 1]$ satisfying $w(v) \geq \epsilon_0$
**Ensure:** Maximum-weight independent set $S^*$
1: $K \leftarrow \lfloor 1/\epsilon_0 \rfloor$                                          ▷ any independent set has size $\leq K$
2: $W_{\text{best}} \leftarrow 0, \; S^* \leftarrow \emptyset$
3: **for all** subsets $X \subseteq V$ with $|X| \leq K$ **do**
4:     **if** $X$ is independent in $G$ **then**
5:         $W \leftarrow \sum_{v \in X} w(v)$
6:         **if** $W > W_{\text{best}}$ **then**
7:             $W_{\text{best}} \leftarrow W, \; S^* \leftarrow X$
8: **return** $S^*$

---

---

**Algorithm 4** TGL-PRUNEDTREE

---

**Require:** Finite-horizon Tree MDP $(\mathcal{S}, \mathcal{A}, P, H)$; Expert-Labeled Trajectory Set $\mathcal{T}_{\text{sel}} = \{\tau_i\}_{i=1}^M$ where $\tau_i = (s_1^i, a_1^i, \ldots, s_H^i, a_H^i)$.
**Ensure:** Pruned set $\mathcal{M}$ with final weights
1: **for all** $\tau_i \in \mathcal{T}_{\text{sel}}$ **do**
2:     $w_i \leftarrow \prod_{h=1}^{H-1} P(s_{h+1}^i \mid s_h^i, a_h^i)$
3: $\mathcal{M} \leftarrow \mathcal{T} = \{(\tau_i, w_i)\}_{i=1}^M, \{(\tau_{\text{com}}, w_{\text{com}})\} \leftarrow \{(\tau_i, w_i)\}$     ▷ active trajectories
4: **for** $h = H, H - 1, \cdots, 1$ **do**                                      ▷ leaf → root
5:     $\texttt{Agg}[(s, a)] \leftarrow 0, \forall (s, a)$                        ▷ map $(s, a) \mapsto$ summed weight
6:     $\texttt{Com}[(s, a)] \leftarrow \varnothing, \forall (s, a)$              ▷ Combine the non-conflicting trajectories
7:     **for each** $(\tau_{\text{com}}, w_{\text{com}}) \in \mathcal{M}$ **do**
8:         Pick any trajectory $\tau \in \tau_{\text{com}}$                     ▷ Just choose a representative
9:         $(s, a) \leftarrow (s_h(\tau), a_h(\tau))$
10:         $\texttt{Agg}[(s, a)] \leftarrow \texttt{Agg}[(s, a)] + w_{\text{com}}$     ▷ Aggregate the weights for the same $(s_h, a_h)$
11:         $\texttt{Com}[(s, a)] \leftarrow \tau_{\text{com}} \cup \texttt{Com}[(s, a)]$
12:     $\mathcal{M} \leftarrow \{(\texttt{Com}[(s, a)], \texttt{Agg}[(s, a)]) | \texttt{Com}[(s, a)] \neq \emptyset\}$
13:     **for each** $s \in \{s | \texttt{Com}[(s, a)] \neq \emptyset\}$ **do**
14:         $a^* \leftarrow \max_{a \in \mathcal{A}} \texttt{Agg}[(s, a)]$
15:         **Delete** $\texttt{Com}[(s, a)], a \neq a^*$ from $\mathcal{M}$
16: $w^* \leftarrow \texttt{Agg}[(s, a)]$
17: **for all** $\tau_i \in \mathcal{M}$ with $\tau_i = (s_1^i, a_1^i, \ldots, s_H^i, a_H^i)$ **do**
18:     **for** $h = 1$ **to** $H$ **do**
19:         $\pi_h(s_h^i) \leftarrow a_h^i$
20: For states not covered by $\bigcup_{v_i \in \mathcal{M}} \tau_i$, set $\pi_h$ via a default rule
21: **return** $\mathcal{M}, \pi$

---

## A.2   Example: BC Fails to Capture Expert Trajectories

**MDP Setup**

- **States:** $S = \{s_1, s_2\}$
- **Actions:** $A = \{a_1, a_2\}$
- **Transitions:** Deterministic transitions from $s_1$ to $s_2$ with any action

**Expert Trajectories**

- $\tau_1 = (s_1, a_1, s_2, a_2)$
- $\tau_2 = (s_1, a_2, s_2, a_1)$

**Behavior Cloning (BC) Limitation** Behavior cloning only observes state-action pairs:

$$(s_1, a_1), \; (s_1, a_2), \; (s_2, a_1), \; (s_2, a_2)$$

resulting in:

- At $s_1$, both $a_1$ and $a_2$ appear equally good.

- At $s_2$, both $a_1$ and $a_2$ appear equally good.

Thus, BC will learn:

$$\pi(a_1|s_1) = \pi(a_2|s_1) = 0.5, \quad \pi(a_1|s_2) = \pi(a_2|s_2) = 0.5$$

**Consequence** Due to this ambiguity, the probability of correctly reproducing either expert trajectory is:

$$P(\tau_1) = \pi(a_1|s_1) \cdot \pi(a_2|s_2) = 0.5 \times 0.5 = 0.25$$

$$P(\tau_2) = \pi(a_2|s_1) \cdot \pi(a_1|s_2) = 0.5 \times 0.5 = 0.25$$

$$\text{Total expert trajectory probability} = 0.5$$

**Direct Trajectory Alignment Advantage** Direct trajectory alignment explicitly optimizes for the sequence:

$$\max_{\pi} \sum_{\tau \in \{\tau_1, \tau_2\}} P_{\pi}(\tau)$$

ensuring the policy preserves the correct sequence with probability 1, with either $\pi(a_1|s_1) = 1, \pi(a_2|s_2) = 1$; or $\pi(a_2|s_1) = 1, \pi(a_1|s_2) = 1$, since it learns to produce both trajectories as whole entities, rather than decomposing them into ambiguous state-action pairs.

### A.3  Proof of Theorem 1

In this section, we want to prove that finding the optimal policy in the binary-labeled expert-labeled trajectory set setting is NP-complete. Before providing the main content of proof. We first define the following variant of MWIS problem.

**Definition 6** (MWIS$_{<1}$)**.**

***Input***

- *A graph $G = (V, E)$;*
- *A weight function $w : V \to (0, 1]$ ;*
- *A rational threshold $K \in (0, 1)$ .*

***Condition*** *For* every *independent set $I \subseteq V$ it holds $\sum\limits_{v \in I} w(v) < 1$.*

***Question*** *Does there exist an independent set $I \subseteq V$ with $\sum\limits_{v \in I} w(v) \geq K$ ?*

**Theorem 5** (MWIS$_{<1}$ is NP–complete)**.** *The decision problem in Definition 6 is NP–complete.*

*Proof.* **Membership in NP.** A certificate is an independent set $I \subseteq V$. We can verify independence in $O(|E|)$ time and compute $\sum_{v \in I} w(v)$ in $O(|I|)$ time, so the problem lies in NP.

We reduce from the classical MAXIMUM INDEPENDENT SET (MIS) problem, known to be NP–complete. The problem is: A graph $G = (V, E)$ and an integer $t \geq 1$, and the question is: does $G$ contain an independent set of size at least $t$?

**Reduction.** Given an instance $(G, t)$ of MIS with $n := |V(G)|$, construct $(G, w, K)$ for MWIS$_{<1}$ as follows

$$w(v) := \frac{1}{n+1} \quad \forall\, v \in V(G), \qquad K := \frac{t}{n+1} \ (< 1).$$

The construction uses only $O(\log n)$-bit rationals, hence is polynomial in $n$.

For *any* independent set $I$ we have

$$\sum_{v \in I} w(v) = \frac{|I|}{n+1} \leq \frac{n}{n+1} < 1,$$

so the promise in Definition 6 is satisfied.

Because every vertex has the same weight,

$$|I| \geq t \iff \sum_{v \in I} w(v) = \frac{|I|}{n+1} \geq \frac{t}{n+1} = K.$$

Thus $(G, t) \in$ MIS iff $(G, w, K) \in$ MWIS$_{<1}$.

The reduction is polynomial, establishing NP–hardness. Since MWIS$_{<1}$ is in NP and NP–hard, it is NP–complete. $\qquad \square$

Now we can establish the proof of Theorem 1.

*Proof.* First, let us restate the original problem.

**Original Problem: Find the optimal deterministic policy**

**Input:** A MDP and the binary expert-labeled trajectory set $\mathcal{T}_{\text{sel}} = \{\tau_i\}_{i=1}^M$ (or called level-1 trajectory)

**Question:** Is there any deterministic policy that can visit the level-1 trajectories with probability at least $p$?

We will do this in two steps:

**Step 1: Show the original problem is in NP**

To show that the original problem is in NP, we must demonstrate that a given solution can be verified in polynomial time.

Given a deterministic policy $\pi = \{\pi_h\}_{h \in [H]}$ , we can:

1. **Check whether the good trajectory does not have conflict with policy $\pi$:**. To be specific, for a good trajectory $\tau = (s_1, a_1, s_2, a_2, \cdots, s_H, a_H)$, check whether $\pi_1(s_1) = a_1, \pi_2(s_2) = a_2, \cdots, \pi_H(s_H) = a_H$. If yes, add up the weight (the probability product) of this trajectory (We denote this set as $Q$). The complexity is at most $O(M \times H)$.

2. **Sum the weights:** Calculate $\sum_{w \in Q} w$ and check if it is at least $p$. This takes at most $O(M)$ time.

Since both checks can be performed in polynomial time, the original problem is in NP.

**Step 2: Reduction from MWIS$_{<1}$**

We will reduce from the MWIS$_{<1}$ problem (Definition 6), which is known to be NP-complete from Theorem 2.

**Reduction from MWIS$_{<1}$ to original problem**

1. **Any weighted graph can be represented as a subset of trajectories with corresponding probability** The basic idea is because we can always find a MDP with sufficient trajectories to represent the relation between these $M$ vertexes, i.e. $M <<$ $O(|S|^H |A|^H)$. The construction process is detailed in Algorithm 5 and 6. To be specific, for an arbitrary graph, we can enumerate every vertex in a Breadth-First Search (**BFS**). Each time we process one vertex and use conflict trajectory pairs to record all the edges it connects to. Then we get a set of trajectories that encoded the information of this graph. The second step is to assign the probability of the MDP given the weights of the graph, which is explained in Algorithm 6. The basic idea is to assign the probability kernel with the given weights $w(v) \in (0, 1]$, and make sure the summation of all probability odds is still 1.

2. **Good policy implies high weight independent set**. First, for any deterministic policy $\pi$, it will produce a set of non-conflicting trajectories (any two of these trajectories are non-conflicting), we denote this set as $\Sigma(\pi)$, and now if a deterministic policy $\pi$ that can visit the level-1 trajectories with probability at least $p$, then $\sum_{w \in \Sigma(\pi)} w > p$, which implies there exists an independent set that the summation of weights is larger than $p$.

Notice that the reduction process in Algorithm 5 and 6 is polynomial in $M$, therefore, we can conclude the original problem is also NP-complete.

**Algorithm 5** BFS-Based Trajectory Construction

---

**Require:** Graph $G = (V, E)$, horizon $H$
**Ensure:** Trajectories $\{\tau(v)\}_{v \in V}$, each of length $H$
 1: Pick an arbitrary root vertex $v_1 \in V$
 2: Fix $\tau(v_1) \leftarrow (s_1, a_1, \ldots, s_H, a_H)$
 3: Visited $\leftarrow \{v_1\}$; enqueue $v_1$ in queue $Q$
 4: **for all** $v \in V \setminus \{v_1\}$ **do**
 5:     $\tau(v) \leftarrow (s_1, a_1, s_U, a_U, \ldots, s_U, a_U)$          ▷ undecided after step 1
 6: **while** $Q$ not empty **do**
 7:     $u \leftarrow \text{Dequeue}(Q)$
 8:     **for all** neighbor $v$ of $u$ **with** $v \notin$ Visited **do**
 9:         Fix remaining undecided steps in $\tau(u)$ so it is unique
10:         Construct $\tau(v)$ so that there exists a time $t$ with $s_t^{\tau(v)} = s_t^{\tau(u)}$ and $a_t^{\tau(v)} \neq a_t^{\tau(u)}$
11:         Visited $\leftarrow$ Visited $\cup \{v\}$; enqueue $v$
12: **return** $\{\tau(v)\}_{v \in V}$

---

**Algorithm 6** Transition-Probability Assignment Using Vertex Weights

---

**Require:** Graph $G = (V, E)$ with weights $w : V \to (0, 1]$, $\sum_{v \in S} w(v) \leq 1$, $S$ is an independent set.
**Require:** Trajectories $\{\tau(v)\}_{v \in V}$ from Alg. 5
**Ensure:** Transition kernel $P(\cdot \mid \cdot, \cdot)$
    /* **Preparation** */
 1: **for all** $v \in V$ **do**
 2:     Write $\tau(v) = (s_1, a_1, \ldots, s_H, a_H)$
    /* **First state–action pair** $(s_1, a_1)$ */
 3: **for all** edge $\{v_1, v_2\} \in E$ **do**
 4:     $P(s_2 \mid s_1, a_1) \leftarrow \max\{w(v_1), w(v_2)\}$
 5:     Distribute remaining mass over other successors of $(s_1, a_1)$
    /* **Finalize each adjacent pair** */
 6: **for all** edge $(u, v) \in E$ **do**
 7:     **if** $w(u) > w(v)$ **then**
 8:         Make the remainder of $\tau(u)$ deterministic
 9:         In $\tau(v)$ add one stochastic step $t$ with prob. $w(v)/w(u)$ where $s_t^{\tau(v)} = s_t^{\tau(u)}$ and $a_t^{\tau(v)} \neq a_t^{\tau(u)}$
10:     **else**
11:         (Symmetric update with $u \leftrightarrow v$)
12: **for all** $(s, a)$ with stochastic successors **do**
13:     Normalize $P(\cdot \mid s, a)$ so $\sum_x P(x \mid s, a) = 1$
14: **return** $P(\cdot \mid \cdot, \cdot)$

---

$\square$

### A.4 Proof of Theorem 2

**Time complexity of Algorithm 1 with oracle Algorithm 3.** Let $M = |\mathcal{T}_{\text{sel}}|$ and $H$ the horizon length, and set

$$K = \left\lfloor \frac{1}{\epsilon_0} \right\rfloor.$$

Then

1. **Conflict-graph construction** (lines 3–7): $O\left(\binom{M}{2} \cdot H\right) = O(M^2 H)$ time.
2. **Weight computation** (lines 9–11): $O(MH)$.
3. **Oracle call** ENUMMWIS$(G, w, \epsilon_0)$ (Alg. 3):

$$\sum_{i=0}^{K} \binom{M}{i} \cdot O(i^2) \;=\; O\left(\sum_{i=0}^{K} \binom{M}{i}\right) \;=\; O\left(M^K\right) \quad \text{(since } K \text{ is a constant)}.$$

4. **Policy extraction** (lines 15–18): $O\left(|S|\,H\right) = O(KH)$.

Putting these together gives

$$T(M, H) \;=\; O\left(M^2 H + MH + M^K + KH\right) \;=\; O\left(M^2 H + M^K\right).$$

In particular, if $\epsilon_0$ (hence $K$) is a fixed constant, this is polynomial time $O(M^2 H + M^{1/\epsilon_0})$.

### A.5 Proof of Theorem 3

*Proof of Time Complexity.* Let:

- $M$ be the number of trajectories,
- $H$ be the trajectory horizon,
- $|\mathcal{A}|$ be the number of discrete actions,
- $|\mathcal{S}|$ be the number of possible states per timestep.

**1. Weight Computation:** Each trajectory $\tau_i$ is assigned a weight via the product of $H - 1$ transition probabilities. Across $M$ trajectories, this requires:

$$\mathcal{O}(M \cdot H)$$

**2. Backward Pruning Loop:** For each timestep $h = H, H - 1, \ldots, 1$ (total $H$ iterations):

- Aggregating weights across identical $(s_h, a_h)$ pairs requires scanning all active trajectories: $\mathcal{O}(M)$,
- Merging trajectories that share the same $(s_h, a_h)$ pair: $\mathcal{O}(M)$,
- For each state $s_h$, selecting the action $a_h$ with the maximum total weight among at most $|\mathcal{A}|$ options leads to: $\mathcal{O}(M \cdot |\mathcal{A}|)$ in the worst case (since there are at most $M$ unique state-action pairs).

Therefore, the total cost per timestep is:

$$\mathcal{O}(M \cdot |\mathcal{A}|)$$

and over $H$ timesteps:

$$\mathcal{O}(H \cdot M \cdot |\mathcal{A}|)$$

**3. Final Output:** The pruned result is returned by traversing at most $M$ remaining elements:

$$\mathcal{O}(M)$$

**Total Time Complexity:** Summing all terms, the dominating component is from the backward pruning loop, yielding the final result:

$$\boxed{\mathcal{O}(H \cdot M \cdot |\mathcal{A}|)}$$

$\square$

## A.6 Proof of Theorem 4

*Proof.* The proof of Theorem 4 is based on the Theorem 1 in [Chen et al., 2013]. For the convenience of the readers, we show the complete process here.

For variable $T_i$, let $T_{i,t}$ be the value of $T_i$ at the end of round $t$, that is, $T_{i,t}$ is the number of times policy played that matches trajectory $v_i$ in the first $t$ rounds. For variable $\hat{\mu}_i$, let $\hat{\mu}_{i,s}$ be the value of $\hat{\mu}_i$ after the policy which matched trajectory $v_i$ is played $s$ times. Then, the value of variable $\hat{\mu}_i$ at the end of round $t$ is $\hat{\mu}_{i,T_{i,t}}$. For variable $u_i$, let $u_{i,t}$ be the value of $u_i$ at the end of round $t$. Let $\overline{\boldsymbol{u}}_t = (u_{1,t}, \ldots, u_{M,t})$ be the random vector fed to the MWIS oracle as the input in line 12 of Algorithm 2 at round $t$.

We also maintain counter $Q_i$ for each trajectory $v_i$ after the $M$ initialization rounds. Let $Q_{i,t}$ be the value of $Q_i$ after the $t$-th round and $Q_{i,M} = 1$. Note that $\sum_i Q_{i,M} = M$. Counters $\{Q_i\}_{i=1}^M$ are updated as follows.

For a round $t > M$, let $S_t$ be the independent set selected in round $t$ by the MWIS oracle (line 12 of Algorithm 2). Round $t$ is bad if the oracle selects a bad set $S_t \in \mathcal{S}_{\text{sub}}$. If round $t$ is bad, let $i = \operatorname{argmin}_{j \in S_t} Q_{j,t-1}$. We increment $Q_i$ by one, i.e., $Q_{i,t} = Q_{i,t-1} + 1$. That is, we find the trajectory $v_i$ with the smallest counter in $S_t$ and increment its counter. If $i$ is not unique, we pick an arbitrary arm with the smallest counter in $S_t$. On the other hand, if $S_t \notin \mathcal{S}_B$, no counter will be incremented.

By definition $Q_{i,t} \le T_{i,t}$. Notice that in every bad round, exactly one counter in $\{Q_i\}_{i=1}^M$ is incremented, so the total number of bad rounds in the first $n$ rounds is less than or equal to $\sum_i Q_{i,n}$.

Define $\ell_t = \frac{6M^2 \ln t}{\Delta_{\min}^2}$. Consider a bad round $t$, $S_t \in \mathcal{S}_{\text{sub}}$ is selected and counter $Q_i$ of some arm $i \in S_t$ is updated. We have

$$\sum_{i=1}^m Q_{i,n} - m \cdot (\ell_n + 1) = \sum_{t=m+1}^n \mathbb{I}\{S_t \in \mathcal{S}_{\text{sub}}\} - m\ell_n$$

$$\le \sum_{t=m+1}^n \sum_{i \in [m]} \mathbb{I}\{S_t \in \mathcal{S}_{\text{sub}}, Q_{i,t} > Q_{i,t-1}, Q_{i,t-1} > \ell_n\}$$

$$\le \sum_{t=m+1}^n \sum_{i \in [m]} \mathbb{I}\{S_t \in \mathcal{S}_{\text{sub}}, Q_{i,t} > Q_{i,t-1}, Q_{i,t-1} > \ell_t\}$$

$$= \sum_{t=m+1}^n \mathbb{I}\{S_t \in \mathcal{S}_{\text{sub}}, \forall i \in S_t, Q_{i,t-1} > \ell_t\}$$

$$\le \sum_{t=m+1}^n \mathbb{I}\{S_t \in \mathcal{S}_{\text{sub}}, \forall i \in S_t, T_{i,t-1} > \ell_t\}$$

We first claim that $\Pr\left(\{S_t \in \mathcal{S}_{\text{sub}}, \forall i \in S_t, T_{i,t-1} > \ell_t\}\right) \le 2 \cdot M \cdot t^{-2}$.

In fact, for any $i \in [M]$,

$$\Pr\left[\left|\hat{\mu}_{i,T_{i,t-1}} - \mu_i\right| \ge \sqrt{3 \ln t / (2 T_{i,t-1})}\right]$$

$$= \sum_{s=1}^{t-1} \Pr\left[\left\{|\hat{\mu}_{i,s} - \mu_i| \ge \sqrt{3 \ln t/(2s)}, T_{i,t-1} = s\right\}\right]$$

$$\le \sum_{s=1}^{t-1} \Pr\left[|\hat{\mu}_{i,s} - \mu_i| \ge \sqrt{3 \ln t/(2s)}\right]$$

$$\le t \cdot 2e^{-3 \ln t} = 2t^{-2}$$

where the last inequality is due to the Chernoff Hoeffding bound. Define $\Lambda_{i,t} = \sqrt{\frac{3 \ln t}{2 T_{i,t-1}}}$ (a random variable since $T_{i,t-1}$ is a random variable), and event $E_t = \left\{\forall i \in [m], \left|\hat{\mu}_{i,T_{i,t-1}} - \mu_i\right| \le \Lambda_{i,t}\right\}$. By union bound, $\Pr\left[\neg E_t\right] \le 2 \cdot M \cdot t^{-2}$. According

to line 11 of Algorithm 2, we have $u_{i,t} - \hat{\mu}_{i,T_{i,t-1}} = \Lambda_{i,t}$. Thus $\left|\hat{\mu}_{i,T_{i,t-1}} - \mu_i\right| \le \Lambda_{i,t}$ implies that $u_{i,t} \ge \mu_i$.

Let $\Lambda = \sqrt{\frac{3\ln t}{2\ell_t}}$, which is not a random variable. Define random variable $\Lambda_t = \max\left\{\Lambda_{i,t} \mid i \in S_t\right\}$. Then

$$E_t \Rightarrow \forall i \in S_t, |u_{i,t} - \mu_i| \le 2\Lambda_t$$
$$\{S_t \in \mathcal{S}_{\text{sub}}, \forall i \in S_t, T_{i,t-1} > \ell_t\} \Rightarrow \Lambda > \Lambda_t$$

Let $\overline{\boldsymbol{u}}_t = (u_{1,t}, \ldots, u_{M,t})$ be the vector representing the adjusted expectation vector at round $t$. Then,

$$E_t \Rightarrow \overline{\boldsymbol{u}}_t \ge \boldsymbol{\mu}$$

If $\{E_t, S_t \in \mathcal{S}_{\text{sub}}, \forall i \in S_t, T_{i,t-1} > \ell_t\}$ holds at time $t$, we have the following important derivation:

$$\sum_{v \in S_t} p_v + 2M\Lambda > \sum_{v \in S_t} p_v + 2M\Lambda_t \ge \sum_{v \in S_t} u_v \ge p^*$$

Since $\ell_t = \frac{6M^2 \ln t}{\Delta_{\min}^2}$, we have $2M\Lambda = \Delta_{\min}$. Therefore,

$$\Pr\left[\{E_t, S_t \in \mathcal{S}_{\text{sub}}, \forall i \in S_t, T_{i,t-1} > \ell_t\}\right] = 0 \Rightarrow$$
$$\Pr\left[\{S_t \in \mathcal{S}_{\text{sub}}, \forall i \in S_t, T_{i,t-1} > \ell_t\}\right]$$
$$\le \Pr\left[\neg E_t\right] \le 2 \cdot M \cdot t^{-2}.$$

The claim thus holds. We have, $\mathbb{E}\left[\sum_{i=1}^M Q_{i,n}\right] \le M(\ell_n + 1) + \sum_{t=1}^n \frac{2M}{t^2} \le \frac{6M^3 \cdot \ln n}{\Delta_{\min}^2} + \left(\frac{\pi^2}{3} + 1\right) \cdot M$.

Notice that each time we select a bad independent set at time $t$, we incur a regret at most $\Delta_{\max}$. Then we obtain the regret bound as follows.

$$\text{Regret}(n)$$
$$\le \mathbb{E}\left[\sum_{i=1}^M Q_{i,n}\right] \cdot \Delta_{\max}$$
$$\le \left(\frac{6\log n \cdot M^2}{\Delta_{\min}^2} + \frac{\pi^2}{3} + 1\right) \cdot M \cdot \Delta_{\max}^2$$
$$= O\left(\frac{M^3 \log K \cdot \Delta_{\max}}{\Delta_{\min}^2}\right)$$

$\square$

## A.7 Experimental Details and Hyperparameters

All hyper-parameters, stopping criteria and environment settings are listed in Table 2.

## A.8 Compute Resources

All runs use a single Google Colab instance with an NVIDIA T4 (16 GB GPU RAM) kernel. Training the PPO expert for 50 000 steps takes around 1 min 20 s; one TGL-UCB run (50 iterations, 20 nodes) takes around 45 s. The evaluation of the three methods that we compare (TGL, BC, and PPO expert) takes around 1 min 30 s. Most of the code are not GPU accelerated, so using a CPU kernel is also feasible.

Table 2: Hyper-parameters used in every experiment.

| Component | Hyper-parameter | Value |
|---|---|---|
| *Environment* | | |
| | Grid size | $4 \times 4$ |
| | Episode horizon $H$ | 10 |
| | Random-action noise | 0.10 |
| *PPO expert* | | |
| | Learning rate | $1 \times 10^{-3}$ |
| | `n_steps` | 128 |
| | Batch size | 64 |
| | Epochs per update | 4 |
| | Discount $\gamma$ | 0.99 |
| | GAE $\lambda$ | 0.95 |
| | Clip range | 0.2 |
| | Total steps | 50 000 |
| *TGL-UCB* | | |
| | $\delta$ | 0.9 |
| | Initial samples $m_0$ | 10 |
| | $\epsilon$ | 0.01 |
| | Max iterations | 50 |
| *Behavioural Cloning* | | |
| | Batch size | 8 |
| | Training epochs | 20 |

