# OpenReview forum: "Trajectory Graph Learning: Aligning with Long Trajectories in Reinforcement Learning Without Reward Design"
_NeurIPS.cc/2025/Conference — NeurIPS 2025 spotlight_

### Official Review · Reviewer_srob · 2025-06-10

**Clarity:** 3
**Significance:** 2
**Originality:** 4
**Rating:** 4
**Confidence:** 3

**Summary:**

The authors present a very interesting and well mathematically analyzed problem formulation to find a optimal subset of expert trajectories that are free of transition conflicts in specific states. The conflict-free trajectories are optimal with respect to the deducable policy that lead to trajctories well aligned to the expert trajectories. They present two algorithms to find policies in environments with unknown or known model dynamics (first is an approximation).

**Questions:**

Is there a difference between (l143) d^phi and (l182) p_i for phi_i? Why is it defined two times?

**Ethical Concerns:**

["NO or VERY MINOR ethics concerns only"]

**Final Justification:**

The authors introduce novel and interesting algorithms. I raise my score to Borderline accept, because they showed during rebuttal, that they address theoretical aspects. However, due to the weaknesses in experiments and writing/style issues, I will not raise to Accept, because these are usually not appropriate to Top-Tier conferences.

**Limitations:**

yes

**Paper Formatting Concerns:**

Nothing

**Quality:**

2

**Strengths And Weaknesses:**

Strengths:
- The mathematical description of the problem is sound and reasonable.
- The proof in showing the np-completeness seems correct and complete.
- The idea of optimizing the MWIS to remove the covariance shift makes sense.

Weaknesses:
- Motivation:
Why should the agent be closer aligned to the expert trajectory with the cost of removing training data? Some removed trajectories could hold valuable information. For example: Was the tweak with "make the holes passable" in the evaluation necessary to make the algorithm able to solve the task?
- Evaluation:
Not reproducable in my opinion. Please explain the experimental setup and evaluation more in detail. For example: what is 'imitation.algorithms.bc'? Is BC and TGL trained on the PPO trajectories?
Also the metric is questionable: Does it make sense that the PPO expert is worse in imitating itself?
Furthermore: Whats the runtime dependency to the problem size? Is it applicable to real-world problems? Maybe the authors could extend the gridsize and measure the runtime.
Abstact says "substantially outperforms standard imitation learning methods for long-trajectory planning." but the experiments only include horizons of 10 steps.


All in all I appreciate the mathematically sound problem formulation and method. I would raise my score to accept, if the authors could give a motivation that explains why the algorithm is useful in practice (maybe with examples) and a more elaborated evaluation that leads to more insights and also presents more metrics like the accuracy on the main tasks.


Minor weaknesses:
- Paper checklist: The instruction block is present, albeit its written to be deleted.
- Some typos.
- L 144 (formula) "Phi in BigPhi" -> BigPhi not introduced.
- Its hard to understand why the problem can be formulated as MWIS and I needed to read it multiple times. Please add a few lines to L 188 that qualitatively describe why a valid policy is equivalent to a independent set in the graph. That would make it easier to follow.

---

> ### Author Rebuttal · Authors · 2025-07-31
>
> We thank the reviewer for recognizing our work's value and providing an excellent summary of our results. Now we address some of the concerns and questions raised by the reviewer.
>
> $\textbf{Q1.}$
> “Motivation: Why should the agent be closer aligned to the expert trajectory with the cost of removing training data? Some removed trajectories could hold valuable information. For example: Was the tweak with "make the holes passable" in the evaluation necessary to make the algorithm able to solve the task?"
>
> $\textbf{A1.}$ Thank you for your question. If we understand correctly, you're asking why our MWIS method selects a subset of expert trajectories—in other words, why we remove some seemingly valuable training data. This is because our objective is to learn a policy that maximizes the probability of matching one of the expert trajectories. While we may have access to a large set of expert demonstrations, many of them may conflict with each other. Intuitively, a single fixed policy cannot simultaneously follow two conflicting trajectories, as discussed in our theoretical analysis. To address this, we employ a graph-based approach to identify a maximum-weight independent set (MWIS), which allows us to select a consistent subset of non-conflicting trajectories for optimizing.
>
> Making the holes passable within our environment was not needed for TGL-UCB algorithm to function, as the TGL algorithm works on the original Frozenlake. The original FrozenLake environment ends an episode when the agent falls into a hole. Making holes passable was just a useful minor modification to allow the failed episodes to continue, which (i) enables longer trajectories for the PPO expert to learn from, (ii) allows for more diverse trajectories and more complex conflicts within our conflict graph structure, and (iii) makes it less trivial to reproduce trajectories. These more diverse and longer (usually full horizon) trajectories are harder to replicate.
>
> $\textbf{Q2.}$
> “Evaluation: Not reproducable in my opinion. Please explain the experimental setup and evaluation more in detail. For example: what is 'imitation.algorithms.bc'? Is BC and TGL trained on the PPO trajectories?"
>
> $\textbf{A2.}$ Thank you for pointing out that the some implementation details were not explicit enough. In our revised appendix and paper, we will add more details such as exact Python packages, seeds, etc., to ensure that our experiment is fully reproducible. Concretely, 'imitation.algorithms.bc' is the behavior-cloning package from the open-source imitation library(https://arxiv.org/abs/2211.11972), which provides the supervised behavior-cloning baseline that we compared against. For the training pipeline, we first train a PPO expert on the modified FrozenLake environment for 50k steps. We roll out this PPO policy to collect a pool of expert demonstration trajectories. TGL and BC both receive this same pool of demonstration trajectories, and no additional information about the environment noise and the PPO expert. The difference is how the two methods (TGL and BC) learn from the demonstration trajectory set. During evaluation, every method is run for 10,000 fresh episodes under the same seed for the environment, and we report the empirical match probability.
>
> $\textbf{Q3.}$
> “Also the metric is questionable: Does it make sense that the PPO expert is worse in imitating itself? Furthermore: What's the runtime dependency to the problem size? Is it applicable to real-world problems? Maybe the authors could extend the gridsize and measure the runtime. Abstact says "substantially outperforms standard imitation learning methods for long-trajectory planning." but the experiments only include horizons of 10 steps."
>
> $\textbf{A3.}$
> "Does it make sense that the PPO expert is worse in imitating itself?": At first glance it may seem counter-intuitive that the PPO expert is not the best at reproducing its own demonstrations, but the gap arises because the objectives are different. PPO is trained to maximize its chances of reaching the goal in the noisy FrozenLake environment, not to reproduce any particular action sequence. When we are sampling trajectories from the PPO expert, we ensured that there are no duplicate trajectories, either through environment noise or to sample from the PPO expert stochastically, which can lead to less efficient detours before reaching the goal. BC and TGL are trained explicitly to replicate the demonstration pool behavior, so they are rewarded for copying the detour behavior. Because our evaluation counts only perfect matches (every state and action must be the same as a demonstration trajectory), PPO's performance is limited to the fraction of demonstrations that lie on its preferred set of paths.
>
>
> Regarding the runtime dependency of our algorithm, we kindly refer the reviewer to Theorems 2 and 4, where the runtime is analyzed in two components: (1) constructing the conflict graph and (2) solving the maximum-weight independent set (MWIS) problem. Both components exhibit polynomial-time complexity with respect to the planning horizon $H$ and the number of expert trajectories $M$. We believe this complexity is reasonable and applicable to certain real-world scenarios.
>
> In response to your suggestion to increase the grid world size, we would like to clarify that the purpose of our experiments is not to demonstrate raw performance in high-dimensional state-action spaces, but rather to isolate and validate the core capability of TGL in solving the trajectory alignment problem using the novel MWIS-based framework. Expanding the grid size to $8 \times 8$ or $10 \times 10$ does increase the number of states, but only by a constant factor. This does not introduce qualitatively new types of trajectory conflicts, nor does it alter the fundamental structure of the conflict graph or the underlying learning dynamics. However, we agree that reporting the runtime could improve transparency, and we are open to including that in future versions for fair comparison.
>
> As for the concern that the planning horizon $H$ is too short, this is primarily due to the limited size of the $4 \times 4$ grid world, which naturally restricts the maximum number of steps. We would also like to clarify a possible misunderstanding regarding our statement in the abstract: when we say our method "substantially outperforms standard imitation learning methods for long-trajectory planning," we are not referring to large values of $H$ per se. Instead, we mean that our method can faithfully imitate entire expert trajectories in a temporally consistent manner—something that standard imitation learning methods often fail to do. The phrase "long-trajectory" here emphasizes exact trajectory alignment over time, rather than the absolute value of $H$.
>
>
> $\textbf{Q4.}$
> “All in all I appreciate the mathematically sound problem formulation and method. I would raise my score to accept, if the authors could give a motivation that explains why the algorithm is useful in practice (maybe with examples) and a more elaborated evaluation that leads to more insights and also presents more metrics like the accuracy on the main tasks."
>
> $\textbf{A4.}$
> Thank you for your appreciation of our theoretical contributions and methodology. Regarding the motivation behind our approach, we offer both analytical and empirical justifications. On the analytical side, we refer the reviewer to Appendix A.2, where we provide concrete examples illustrating how behavior cloning (BC) can fail to capture expert trajectories. The key insight is that traditional methods like BC often rely solely on cumulative rewards and overlook the sequential structure of trajectories. In contrast, our method directly aligns with complete trajectories, thereby capturing richer temporal information. This leads to improved performance on trajectory-level metrics such as matching probability.
>
> On the empirical side, we draw motivation from real-world applications—such as self-driving and large language models (LLMs)—where decision sequences are long, branching, and structurally complex. For example, in self-driving, trajectory planning involves reasoning over distinct, non-overlapping future paths; in LLMs, decoding strategies like beam search produce diverse, tree-structured output sequences. These settings highlight the need to model full trajectories rather than rely solely on scalar feedback.
>
> As for your suggestion regarding more detailed evaluations and metrics, we clarify that our work introduces a new setting—direct trajectory alignment—where the natural evaluation criterion is the probability that a learned policy exactly reproduces the expert trajectory set. Since the goal is replication rather than reward maximization, traditional RL metrics such as episode return or proximity to a ground-truth policy can be uninformative or even misleading. For example, a policy may achieve high returns while never matching any expert trajectories. In our framework, policy quality is judged solely by its alignment with the provided demonstrations, which is why our experiments and metrics are designed around this alignment objective.
>
> $\textbf{Q5.}$
> “Minor weaknesses"
>
> $\textbf{A5.}$
> Thank you for pointing out these minor issues, and we will correct them in a revised version.
>
> $\textbf{Q6.}$
> “Is there a difference between (l143) $d^\pi$ and (l182) $p_i$ for $\tau_i$? Why is it defined two times?"
>
> $\textbf{A6.}$
> Thank you for your question. In line 143, $d^\pi(\tau)$ denotes the probability of generating trajectory $\tau$ under policy $\pi$—that is, it incorporates both the policy's action distribution and the environment's transition dynamics. In contrast, in line 182, $p_i$ refers to the raw probability of trajectory $\tau_i$ based solely on the product of transition probabilities. These two quantities differ in that $d^\pi(\tau)$ accounts for the stochasticity of the policy, whereas $p_i$ considers only the environment's dynamics.

---

> ### Comment · Reviewer_srob · 2025-08-01
>
> Thank you for your rebuttal. I will raise my score to Borderline Accept.

---

> > ### Author Response · Authors · 2025-08-01
> >
> > We’re truly grateful that you recognized the novelty and value of our work, and heartened that our detailed rebuttal addressed your concerns—thank you for raising your recommendation to accept, which is especially encouraging and affirming as we move forward.

---

### Official Review · Reviewer_nNgV · 2025-06-27

**Clarity:** 3
**Significance:** 2
**Originality:** 3
**Rating:** 4
**Confidence:** 4

**Summary:**

This paper investigates the direct trajectory alignment problem, which aims to find a deterministic policy that maximizes the visitation probability of a given set of expert trajectories. The authors make a key theoretical contribution by proving that this problem is NP-complete, by showing its equivalence to the maximum-weight independent set problem. To overcome this computational hardness, the paper proposes polynomial-time algorithms for two special cases under known dynamics. For the unknown dynamics, a UCB-based online learning algorithm is developed and analyzed.

**Questions:**

- In general, the use of direct trajectory alignment is not well motivated. In the example in Appendix A.2, it is unclear to me why the DTA policy is superior than the BC policy. Could you elaborate on when and why DTA is better than BC?

- In numerical experiments, could you present other performance metrics (e.g. distance to a ground-truth expert policy or task performances of the resulting policy). Additionally, how does the policy's performance vary as the horizon length (H) changes?

- In line 240, I believe the complexity bound should be M^{1/\epsilon_0}.

**Ethical Concerns:**

["NO or VERY MINOR ethics concerns only"]

**Final Justification:**

I have increased my score, as the authors have addressed my concerns.

**Limitations:**

yes

**Paper Formatting Concerns:**

I think the abstract should be a single paragraph.

**Quality:**

2

**Strengths And Weaknesses:**

Strengths:
- The proposed framework of direct trajectory alignment is a novel and interesting contribution, as it bypasses the need for reward model or any model assumptions.
- The paper is well-structured and clearly presents its problem formulation, algorithms, and theoretical analyses.

Weaknesses:
- The proposed approaches in Section 4 and 5 appear to be well established techniques and their error bounds are also naïve.
- While the paper claims that the method is effective for long-horizon problems, but this central claim lacks sufficient analytical or empirical support. It is not made clear why optimizing the proposed objective function inherently leads to better performance on long-horizon tasks compared to other methods.
- The experimental validation is limited. The evaluation is conducted on a single, small-scale grid-world environment and the metric is the trajectory match probability, which is directly related to the method's objective function. This makes it difficult to assess the framework's generalization capabilities or its effectiveness on other standard performance metrics.
- The analysis is limited to deterministic policies and the tabular setting.

---

> ### Author Rebuttal · Authors · 2025-07-31
>
> We thank the reviewer for recognizing our work's value and providing an excellent summary of our results. Now we address some of the concerns and questions raised by the reviewer.
>
> $\textbf{Q1.}$
> “The proposed approaches in Section 4 and 5 appear to be well established techniques and their error bounds are also naïve."
>
> $\textbf{A1.}$
> Thank you for your question. In Section 4, we introduce two practically motivated settings: bounded realizability of the expert trajectory set and tree-structured MDPs. Both settings are grounded in real-world scenarios and, importantly, admit polynomial-time algorithms. While the theoretical bounds we provide may appear relatively simple, the key contribution lies in identifying meaningful and realistic cases in which the otherwise NP-hard MWIS problem becomes tractable. In Section 5, we further develop a novel UCB-style exploration strategy tailored to our TGL framework. To analyze its sample complexity, we reduce the problem to a combinatorial multi-armed bandit (CMAB) setting. This reduction is non-trivial and reflects a careful design choice to make the exploration problem theoretically analyzable while preserving the core structure of trajectory alignment.
>
> $\textbf{Q2.}$
> “While the paper claims that the method is effective for long-horizon problems, but this central claim lacks sufficient analytical or empirical support. It is not made clear why optimizing the proposed objective function inherently leads to better performance on long-horizon tasks compared to other methods."
>
> $\textbf{A2.}$
> Thank you for raising the concern regarding the analytical and empirical support for our method’s effectiveness in long-horizon problems. For analytical support, we kindly refer the reviewer to Appendix A.2, where we provide concrete examples illustrating how behavior cloning (BC) fails to capture expert trajectories. The key intuition is that traditional methods, such as BC, often rely solely on cumulative rewards and fail to preserve the sequential structure of trajectories. In contrast, our method directly aligns with entire trajectories, capturing richer and more complete information. This leads to better performance under trajectory-level metrics such as matching probability. On the empirical side, we have also discussed practical motivations in real-world applications—such as self-driving and large language models (LLMs)—where decision sequences are long and branch significantly. For instance, in self-driving, trajectory planning involves reasoning over long, non-overlapping future paths; in LLMs, decoding strategies like beam search generate diverse sequences that resemble tree-structured decisions. These examples further highlight the importance of modeling full trajectories rather than relying on scalar feedback alone.
>
>
> $\textbf{Q3.}$
> “The experimental validation is limited. The evaluation is conducted on a single, small-scale grid-world environment and the metric is the trajectory match probability, which is directly related to the method's objective function. This makes it difficult to assess the framework's generalization capabilities or its effectiveness on other standard performance metrics."
>
> $\textbf{A3.}$
> Thank you for highlighting the scope of our experiment. Our experiment deliberately focuses on a small, discrete grid-world environment because (i) it allows us to control for confounding factors and isolate the effect of our new trajectory-alignment framework and objective, and (ii) the theoretical guarantees we have proven currently assume a finite state-action space, so extending the experiments beyond finite MDPs first requires extending our theoretical work. While the evaluation metric, trajectory match probability, indeed relates to the objective function, this similarity is intentional. In the direct trajectory alignment setting we propose, a policy is only considered "good" when it reproduces the demonstrations, independent of learned or assigned rewards. For example, even if the demonstration trajectory set is objectively "poor" based on rewards, the desired goal of our framework remains the same: match the demonstrations. That said, we agree that supplementing match probability with other metrics will help assess the framework's generalization capabilities. In the revised version of our paper, we will add additional measurements of trajectory quality and conflict-graph density, and error bars as a statistical measure to show the applicability of our framework in larger environments.
>
> $\textbf{Q4.}$
> “The analysis is limited to deterministic policies and the tabular setting."
>
> $\textbf{A4.}$
> Thank you for the thoughtful suggestion. First, our paper begins with the deterministic policy setting because it offers a clearer formulation and is more straightforward to analyze. However, extending to stochastic policies is not a fundamental issue in our framework. In fact, for the trajectory alignment problem we consider, the optimal policy can always be found within the set of deterministic policies. One can think of the space of all policies as a polyhedron, where deterministic policies correspond to its vertices. Since the optimization objective is linear with respect to the policy distribution, the optimal solution lies on the boundary of this polyhedron—i.e., at a deterministic policy. Thus, it suffices to focus on deterministic policies.
>
> Second, while our setting assumes a finite action space to ensure polynomial-time tractability, the state space can be infinite. This is because the core computational complexity of the TGL framework arises from learning the trajectory alignment graph, which depends on the number of expert trajectories rather than the total number of states. As a result, in the sample complexity bound of our UCB-style exploration method (Section 5), the number of states does not appear. This shows that our method is not restricted to the tabular setting and can generalize to environments with large or even infinite state spaces.
>
> $\textbf{Q5.}$
> “In general, the use of direct trajectory alignment is not well motivated. In the example in Appendix A.2, it is unclear to me why the DTA policy is superior than the BC policy. Could you elaborate on when and why DTA is better than BC?"
>
> $\textbf{A5.}$
> Thank you for your question regarding the example in Appendix A.2. We are happy to provide a more detailed explanation. In this example, we consider two expert trajectories to align with: $\tau_1 = (s_1, a_1, s_2, a_2)$ and $\tau_2 = (s_1, a_2, s_2, a_1)$. Traditional behavior cloning (BC) treats each state-action pair independently. In this case, since both $a_1$ and $a_2$ appear equally often at both $s_1$ and $s_2$ across the two trajectories, BC learns a uniform policy:
>
> $$
> \pi_1(a_1 \mid s_1) = \pi_1(a_2 \mid s_1) = 0.5,\quad \pi_2(a_1 \mid s_2) = \pi_2(a_2 \mid s_2) = 0.5.
> $$
>
> where $\pi_1$ and $\pi_2$ are the policies in step 1 and 2 respectively.
> However, this policy fails to capture the sequential structure of either trajectory. The probability of reproducing each expert trajectory under this policy is:
>
> $$
> P(\tau_1) = \pi_1(a_1 \mid s_1) \cdot \pi_2(a_2 \mid s_2) = 0.5 \times 0.5 = 0.25,
> $$
> $$
> P(\tau_2) = \pi_1(a_2 \mid s_1) \cdot \pi_2(a_1 \mid s_2) = 0.5 \times 0.5 = 0.25,
> $$
>
> so the total matching probability for expert trajectories is only:
> $$
> P(\tau_1) + P(\tau_2) = 0.25 + 0.25 = 0.5.
> $$
>
> In contrast, Direct Trajectory Alignment explicitly optimizes the sequence-level probability:
>
> $$
> \max_{\pi} \sum_{\tau \in \{\tau_1, \tau_2\}} P_\pi(\tau),
> $$
>
> allowing the policy to recover the full structure of either trajectory. For instance, setting $\pi_1(a_1 \mid s_1) = 1$, $\pi_2(a_2 \mid s_2) = 1$ will reproduce $\tau_1$ with probability 1; similarly, $\pi_1(a_2 \mid s_1) = 1$, $\pi_2(a_1 \mid s_2) = 1$ will reproduce $\tau_2$. In both cases, the total expert trajectory
>
> $\textbf{Q6.}$
> “In numerical experiments, could you present other performance metrics (e.g. distance to a ground-truth expert policy or task performances of the resulting policy). Additionally, how does the policy's performance vary as the horizon length (H) changes?"
>
> $\textbf{A6.}$ We thank the reviewer for the suggestion, but would like to clarify that our work introduces a new setting, direct trajectory alignment, for which the natural evaluation criterion is the probability that the resulting policy reproduces the finite expert trajectory set. Because the goal is replication rather than reward maximization, traditional RL metrics such as episode return or distance to an optimal "ground-truth" policy are not informative and could even be misleading. For example, a policy may score highly on those metrics while never matching any demonstration trajectories. In our framework, how good our policy is is determined solely by the quality and diversity of the provided trajectory set, which is why we designed our experiment in this way. As for the second question, by construction, the achievable match probability to the trajectory set does not change when horizon (H) is changed, since the number of trajectories remains fixed. As we change H, the runtime complexity of the resampling operation within our algorithm will scale linearly.
>
> $\textbf{Q7.}$ In line 240, I believe the complexity bound should be $M^{1/\epsilon_0}$.
>
> $\textbf{A7.}$
> Thank you for pointing out the typo. It should be $M^{1/\varepsilon_0}$.

---

> > ### Comment · Reviewer_nNgV · 2025-08-04
> >
> > Thank you for the clarifications and detailed responses. I have two additional questions to authors:
> >
> > (i) The DTA objective is interesting, but the motivation as presented is still not very convincing.
> > For example, in the given example, why should the policy produce one of the expert trajectories, $\tau_1$ or $\tau_2$?
> > Is this because the trajectories not included in the expert demonstration dataset but still producible by a BC policy ($(s_1, a_1, s_2, a_1)$ and $(s_1, a_2, s_2, a_2)$ in this case) might be toxic or unsafe in practice?
> >
> > (ii) In the experiments, TGL policies are shown to achieve a high trajectory match probability, which is expected since the policies are optimized for this objective. However, the generalization capabilities of TGL policies remain unclear. Could you comment on what TGL potentially sacrifices in achieving high trajectory match probability?

---

> > > ### Author Response · Authors · 2025-08-04
> > >
> > > Thank you for your follow‑up questions regarding the motivation and scope of TGL.
> > >
> > > (i) You’re absolutely right about the fact that trajectories that fall outside expert demonstrations can be unsafe or even toxic, despite achieving high reward under standard reinforcement‑learning frameworks. This underscores a key limitation of traditional reward models and motivates TGL’s focus on strict, full‑trajectory alignment with expert examples.
> > >
> > > For instance, in large‑language‑model fine‑tuning, reward models that score at the token or word level fail to capture the holistic meaning of a sentence—changing even a single word can dramatically alter its intent or effectiveness. TGL evaluates complete responses, avoiding fragmented token‑level rewards and preserving nuance. Similarly, in autonomous driving, expert trajectories often include delicate maneuvers — e.g., steering through narrow roads or tunnels—where minor deviations can lead to dangerous outcomes. Traditional reward functions frequently miss these subtle but critical differences, whereas TGL naturally encodes expert judgment across entire trajectories, faithfully reflecting demonstration intent.
> > >
> > > In the simple example you’re interested in: behavioral cloning (BC) treats $ \((s_1, a_1)\) $ and $ \((s_1, a_2)\) $ as equivalent if they appear equally often in expert data, entirely ignoring the sequence $ \((s_1, a_1, s_2, a_2)\) $. The consequence is that BC‑trained policies may respond to dialogue without respecting earlier context, or in self‑driving tasks, they may “forget” that the car is navigating a narrow road or tunnel—leading to hazardous steering decisions and potential crashes. This limitation stems from BC being a supervised, state–action regression method trained on i.i.d. samples, with no inherent awareness of temporal structure.
> > >
> > > (ii) TGL is built to maximize the probability of reproducing the demonstrated trajectories. As a consequence, TGL does not attempt to optimize an inferred reward, nor does it extrapolate to unseen parts of the state space. In environments where the agent must act far beyond the demonstrated states and actions, this focus can become a limitation. Additionally, since TGL must solve a MWIS problem on the trajectory conflict graph, the runtime complexity becomes a bottleneck when we have complex trajectory conflicts, which is why we suggested several practically motivated graph structures (bounded-realizable sets, tree structure) that allow polynomial-time solutions. Conversely, in domains where staying on the demonstration manifold is essential (LLM finetuning, autonomous driving, etc.), this trajectory alignment behavior is a core advantage, and not a drawback. In these scenarios, TGL ensures that generated trajectories remain within the bounds of human-approved behavior. Thus, while TGL sacrifices worst-case runtime complexity compared to simpler baselines, it gains the ability to maximize the Direct Trajectory Alignment (DTA) objective, which is an objective that standard reward-seeking methods (IRL, PbRL, etc.) or per-state imitation methods (BC) cannot guarantee.
> > >
> > > We hope this addresses both the motivation for strict trajectory alignment and the boundary of TGL's framework.

---

> > > > ### Comment · Reviewer_nNgV · 2025-08-04
> > > >
> > > > Thank you for the responses. I generally agree with your points and encourage you to address them in the revised manuscript. I will raise my score to a borderline accept.

---

> > > > > ### Author Response · Authors · 2025-08-04
> > > > >
> > > > > Thank you for taking the time to review our response and confirming that our clarifications have resolved your concerns. We’re especially grateful that you’ve decided to raise your overall score - your support means a great deal.

---

### Official Review · Reviewer_CpAw · 2025-06-30

**Clarity:** 2
**Significance:** 3
**Originality:** 3
**Rating:** 4
**Confidence:** 2

**Summary:**

The paper explores the problem of directly aligning policies with expert-labeled trajectories in reinforcement learning without relying on reward modeling. It proposes Trajectory Graph Learning (TGL), a framework that leverages structural assumptions to enable efficient policy planning.

**Questions:**

1. Given the current experiments are only conducted on a 4×4 Frozen Lake environment, it would be beneficial to validate the TGL framework on more complex environments such as larger grid-worlds or continuous state/action spaces. For example, environments like the Mujoco robotics simulations or more complex navigation tasks could provide a better understanding of the framework’s generalizability and robustness.
2. Could you include comparisons with related methods such as Inverse RL and PbRL in the experiments? This would help to demonstrate the practical utility and advantages of the TGL framework beyond its theoretical merits.

**Ethical Concerns:**

["NO or VERY MINOR ethics concerns only"]

**Final Justification:**

• Addressed issues

– Authors will clearly state the paper’s scope (discrete, finite MDPs) and add a dedicated “Limitations & Future Work” section that explicitly discusses the need for broader benchmarks, scalability, and extension to continuous or high-dimensional domains.

– Reproducibility and formatting concerns (supplementary ZIP) are acknowledged and will be corrected.


• Unresolved issues

– No new empirical results beyond the 4×4 Frozen Lake; continuous or larger-scale validation remains future work.


• Reason for raising score to 4 (Borderline Accept)

– The core contribution is a well-posed theoretical formulation and first algorithm for Direct Trajectory Alignment, accompanied by proofs of hardness and regret guarantees. The community traditionally values such foundational theory even when immediate practical impact is limited. After rebuttal, the authors commit to clarifying limitations and future directions, satisfying my main concerns.

**Limitations:**

The authors have not adequately addressed the limitations of their work. It is suggested to supplement the discussion on the limitations of the TGL framework in the paper, such as its applicability in different tasks and environments.

**Paper Formatting Concerns:**

The main text contains a reference to supplementary material, but the supplementary material is missing is provided as a single ZIP file containing the full paper with appendices. However, including the full paper in the ZIP file is not in line with the guidelines.

**Quality:**

2

**Strengths And Weaknesses:**

* Strengths:
1. The motivation of the paper is strong. It highlights the limitation of traditional behavior cloning in imitation learning, which only focuses on state-action pair alignment and neglects the broader trajectory-level structure, and proposes the idea of direct trajectory alignment, which is meaningful.
2. The proposal of the TGL framework is innovative to some extent, aiming to solve the problem of trajectory alignment under structural assumptions.
* Weaknesses:
1. The paper is relatively theoretical, and some details are not well explained. For instance, the supplementary material includes the full paper with appendices in a single ZIP file, which is not in line with the NeurIPS submission guidelines. Reviewers primarily rely on the officially submitted PDF for their assessment, and any additional technical appendices should ideally be submitted together with the main paper before the full submission deadline to facilitate a more straightforward review process.
2. The theoretical part of the paper is relatively strong, but to enhance the practical significance of the research, it would be valuable to include comparisons with related methods such as Inverse RL and PbRL in the experiments. This would help to demonstrate the practical utility and advantages of the TGL framework beyond its theoretical merits.
3. Given the current experiments are only conducted on a 4×4 Frozen Lake environment, it would be beneficial to validate the TGL framework on more complex environments such as larger grid-worlds or continuous state/action spaces. For example, environments like the Mujoco robotics simulations or more complex navigation tasks could provide a better understanding of the framework’s generalizability and robustness.

---

> ### Author Rebuttal · Authors · 2025-07-31
>
> We thank the reviewer for recognizing the value of our work and providing valuable feedback. We address the concerns of the reviewer as follows.
>
> $\textbf{Q1.}$
> “The paper is relatively theoretical, and some details are not well explained. For instance, the supplementary material includes the full paper with appendices in a single ZIP file, which is not in line with the NeurIPS submission guidelines. Reviewers primarily rely on the officially submitted PDF for their assessment, and any additional technical appendices should ideally be submitted together with the main paper before the full submission deadline to facilitate a more straightforward review process."
>
> $\textbf{A1.}$ We thank the reviewer for pointing this out and sincerely apologize for not adhering strictly to the NeurIPS submission guidelines regarding the formatting and inclusion of supplementary materials. We acknowledge that submitting the full paper and appendices as a single ZIP file may have made it more difficult to access key technical details during the review process. We will ensure that all technical appendices are properly integrated into the main supplementary PDF in accordance with the official requirements in any future submissions or revisions. We appreciate the reviewer’s understanding and will take greater care to streamline the presentation for ease of review.
>
>
> $\textbf{Q2.}$
> “The theoretical part of the paper is relatively strong, but to enhance the practical significance of the research, it would be valuable to include comparisons with related methods such as Inverse RL and PbRL in the experiments. This would help to demonstrate the practical utility and advantages of the TGL framework beyond its theoretical merits."
>
> $\textbf{A2.}$
> We appreciate the reviewer’s recognition of the theoretical strength of our paper. Regarding the suggestion to include comparisons with related methods such as Inverse Reinforcement Learning (IRL) and Preference-based Reinforcement Learning (PbRL), we would like to clarify that although these methods are discussed in the motivation and related work sections, our proposed TGL framework addresses a fundamentally different problem—namely, the General Direct Trajectory Alignment problem. Traditional IRL and PbRL approaches typically aim to recover a reward function and evaluate performance through value-based metrics. In contrast, TGL evaluates performance by measuring the matching probability of expert trajectories, which aligns more closely with the objectives of behavior cloning methods. This difference in problem formulation and evaluation criterion is the main reason why we chose to compare TGL against behavior cloning baselines in our experiments.
>
>
>
> $\textbf{Q3.}$
> “Given the current experiments are only conducted on a 4×4 Frozen Lake environment, it would be beneficial to validate the TGL framework on more complex environments such as larger grid-worlds or continuous state/action spaces. For example, environments like the Mujoco robotics simulations or more complex navigation tasks could provide a better understanding of the framework’s generalizability and robustness."
>
>
> $\textbf{A3.}$ We appreciate the reviewer’s suggestion to evaluate our proposed TGL framework in more complex and real-world environments to further assess its scalability and robustness. We would like to clarify that the theoretical foundation of TGL is built on a Markov Decision Process (MDP) with a finite number of states and actions, as well as a finite set of expert trajectories. As such, the current framework does not extend to continuous MDPs or settings involving infinitely many expert trajectories. With respect to larger environments, our experimental objective is not to highlight raw performance in high-dimensional state and action spaces, but rather to isolate and validate TGL’s effectiveness in solving the trajectory alignment problem through our novel MWIS-based method. Scaling the grid to larger sizes—such as $8 \times 8$ or $10 \times 10$—increases the number of states by a constant factor, but does not introduce qualitatively new types of trajectory conflicts. Consequently, the underlying graph structure and learning dynamics of TGL remain unchanged.
>
> $\textbf{Q4.}$ “Given the current experiments are only conducted on a 4×4 Frozen Lake environment, it would be beneficial to validate the TGL framework on more complex environments such as larger grid-worlds or continuous state/action spaces. For example, environments like the Mujoco robotics simulations or more complex navigation tasks could provide a better understanding of the framework’s generalizability and robustness."
>
> $\textbf{A4.}$  We kindly refer the reviewer to our response to Question 3 (Answer 3), where we have addressed this point in detail.
>
> $\textbf{Q5.}$ “Could you include comparisons with related methods such as Inverse RL and PbRL in the experiments? This would help to demonstrate the practical utility and advantages of the TGL framework beyond its theoretical merits."
>
> $\textbf{A5.}$  We kindly refer the reviewer to our response to Question 2 (Answer 2), where we have addressed this point in detail.

---

> > ### Comment · Reviewer_CpAw · 2025-08-01
> >
> > Thank you for the detailed rebuttal. While aligning with expert trajectories is well-motivated, the ultimate value of any policy lies in its ability to solve downstream tasks effectively. Evidence that TGL outperforms BC, IRL, PbRL or other RL baselines on standard benchmarks (e.g., MuJoCo control, Atari, or larger navigation tasks) is still missing. Without such comparisons, the current evaluation risks being a theoretical exercise in a toy domain. I encourage you to include these task-performance results to demonstrate practical impact.

---

> > > ### Author Response · Authors · 2025-08-01
> > >
> > > We thank the reviewer for the constructive follow-up and welcome the opportunity to clarify. As noted in A2 and A3 of our rebuttal, TGL addresses direct trajectory alignment, which is a setting in which success is defined by $\textit{faithfully}$ reproducing the provided demonstrations rather than by maximizing an external or inferred reward. Hence, our evaluation focuses on trajectory match probability, the quantity most aligned with our theory, and comparisons with behavioral cloning, a baseline that shares the same objective.
> > >
> > > Reward-seeking baselines such as IRL, PbRL, and standard RL aim to optimize an (external or inferred) reward function, whereas TGL’s objective is to replicate the demonstrations exactly, regardless of whether those demonstrations are high- or low-reward. Comparing the two would therefore conflate the two questions: “how can we infer rewards about the environment and maximize the rewards?” and “how faithfully can we reproduce the demonstrations?”.
> > >
> > > Regarding the environments you mention: MuJoCo control tasks are continuous-state MDPs, so applying TGL there first requires extending our discrete-state theory (an open direction we consider as future work). Atari, while discrete in principle, represents each state as a high-dimensional RGB frame; exact state repeats across trajectories are virtually nonexistent, leaving no trajectory conflicts and reducing TGL (MWIS of the conflict graph) into a trivial union of the demonstration trajectories. Therefore, neither domain offers a meaningful test of the trajectory-conflict mechanisms our method is designed to handle.
> > >
> > > Within the tabular scope covered by our theoretical analysis, we already observe that TGL consistently achieves higher match probability than BC and a PPO expert across several deterministic/stochastic trajectory splits, underscoring the practical value of the framework even before these broader extensions.

---

> > > > ### Comment · Reviewer_CpAw · 2025-08-02
> > > >
> > > > Thank you for the clarifications.
> > > > Direct trajectory alignment is indeed a worthwhile goal, but a method becomes valuable only after the community agrees on *why* and *how* it should be measured. Rather than retrofitting a new metric to fit TGL, the paper should first establish a benchmark that (i) covers both toy and realistic domains (autonomous driving, LLM decoding, etc.), (ii) defines new evaluation criteria that practitioners judge as meaningful, and (iii) provides baselines (BC, IRL, PbRL, RL) under these criteria. Once such a benchmark is accepted, demonstrating that TGL tops it would make a complete contribution. As it stands, the work feels “solution-first.” I recommend restructuring the paper around the benchmark creation and validation effort, then situating TGL as the first strong baseline within that framework.

---

> > > > > ### Author Response · Authors · 2025-08-04
> > > > >
> > > > > Thank you for the thoughtful follow-up. We agree that a broadly accepted benchmark suite for Direct Trajectory Alignment (DTA) would benefit the community, and we share that long-term goal. Our present paper tackles the foundational step: we formalize DTA as a new objective, prove its computational hardness, and introduce the first tractable algorithms (TGL) with supporting experiments. In this setting, empirical trajectory-match probability is the cleanest and most direct measure of alignment with the expert set, similar to exact-match/log-likelihood in sequence modeling, because the objective itself is to reproduce the demonstrations.
> > > > >
> > > > > We also recognize that, in practice, DTA may be paired with domain-specific criteria (e.g., safety/comfort in autonomous driving, human-preference signals in language). Designing such benchmarks entails curating new datasets and defining application-level metrics that vary by domain, which is beyond the scope of this paper. Our aim here is to lay the groundwork. Within the discrete, tabular scope covered by our analysis, the $ 4 \times 4 $ Frozen-Lake study captures the core mechanism we analyze, which is the trajectory conflicts and their resolution via the conflict graph; enlarging the grid (e.g., $ 8 \times 8 $, $ 10 \times 10 $) increases the number of states but does not introduce qualitatively new conflict structures, so the graph-learning dynamics of TGL remain the same. By showing that TGL consistently achieves higher match probability than BC (which shares a similar objective) in this controlled yet representative environment, we were able to validate the key ideas within our framework and algorithms.
> > > > >
> > > > > In short, we consider our paper as “problem-first”: we introduce a novel, well-motivated objective, supply the mathematics and algorithms, and demonstrate their behavior in a controlled environment. Our primary contributions is the theoretical foundation and analysis. The empirical study serves mainly to validate our theory, and to lay the foundation for broader benchmark efforts.

---

> > > > > > ### Comment · Reviewer_CpAw · 2025-08-05
> > > > > >
> > > > > > Thank you for your response. I encourage you to incorporate the discussion on the limitations and future work into the paper, highlighting the need for broader benchmarking and domain-specific criteria. I will carefully consider your points and engage in further discussion with other reviewers and AC.

---

> > > > > > > ### Author Response · Authors · 2025-08-05
> > > > > > >
> > > > > > > Thank you for your thoughtful response and for encouraging us to elaborate on the limitations and future directions. We will revise the manuscript to include a dedicated section that addresses the need for broader benchmarking and domain-specific evaluation criteria. We appreciate your engagement and look forward to further discussion.

---

> > > > > > > > ### Author Response · Authors · 2025-08-07
> > > > > > > >
> > > > > > > > Thank you for your feedback on expanding the discussion of limitations and future work in our paper. We find this feedback very valuable, as it will help the community better understand and further explore the novel ideas introduced in our work. As discussed in our previous comment, we will add the following descriptions into the revised version of our paper to address the points we discussed. Finally, we would be happy to discuss further if you have any additional questions or suggestions.
> > > > > > > >
> > > > > > > > $\textbf{Limitations}$. Our primary contribution in this work is the theoretical foundation and analysis of the direct trajectory alignment problem. We establish computational hardness results, propose the Trajectory Graph Learning (TGL) framework, and provide regret guarantees under unknown dynamics. The empirical study serves mainly to validate these theoretical contributions and is limited to a single $4\times4$ Frozen Lake environment. Scaling the grid to larger sizes—such as $ 8 \times 8$ or $ 10 \times10 $ increases the number of states but does not introduce qualitatively new conflict structures, so the graph-learning dynamics of TGL remain the same. Additionally, while TGL avoids reward modeling, it still depends on a curated set of expert trajectories, which may need to be carefully designed in practice to reflect domain-specific goals or safety criteria. Computationally, the TGL-UCB algorithm is more expensive in the worst case than traditional methods like behavioral cloning, and scalability remains a concern as problem size grows.
> > > > > > > >
> > > > > > > > $\textbf{Future Work}$. A key direction going forward is extending the theory to continuous MDP settings and the development of trajectory-level RL benchmarks, specifically designed to evaluate alignment without reward modeling. This may involve adapting existing environments or designing new ones where trajectory sets can be meaningfully constructed, compared, and evaluated. Building such benchmarks is non-trivial and will likely require substantial engineering effort, especially in continuous or high-dimensional domains such as robotics, autonomous driving, or natural language generation. Additionally, improving the efficiency of TGL, through approximate solvers, amortized inference, or scalable graph-based methods, will be crucial for extending its applicability to larger and more complex settings.

---

### Official Review · Reviewer_1Der · 2025-07-02

**Clarity:** 3
**Significance:** 3
**Originality:** 3
**Rating:** 4
**Confidence:** 3

**Summary:**

This paper proposes Trajectory Graph Learning (TGL), a novel framework that directly aligns policies with expert-labeled trajectories to preserve long-horizon behavior without relying on reward signals. By modeling the trajectory alignment problem as a Maximum-Weight Independent Set (MWIS) problem, the authors address the inherent complexity of the problem and develop efficient algorithms for both known and unknown environment dynamics. The theoretical analyses demonstrate the NP-completeness of the general trajectory alignment problem and provide polynomial-time solutions under certain structural assumptions. Experimental results on grid-world tasks show that TGL outperforms standard imitation learning methods, especially for long-trajectory planning.

**Questions:**

1.	Can the authors conduct experiments on more complex and real-world environments to further validate the scalability and robustness of the proposed TGL framework?

2.	Can the authors report error bars or perform statistical significance tests on their experimental results to provide a more rigorous evaluation of the performance of TGL compared to baselines?

3.	Can the authors provide more details on the practical relevance and validity of the structural assumptions (bounded realizability of expert trajectories, tree-structured MDP) made in their theoretical analyses?

**Ethical Concerns:**

["NO or VERY MINOR ethics concerns only"]

**Final Justification:**

After considering the author response and the other reviews, I find my concerns have been adequately addressed. Specifically:

+ The authors clarified why the method is not applicable to continuous MDP tasks, which appropriately bounds its scope.

+ They added error bars to the experimental results and strengthened the discussion of their theoretical assumptions.

+ They committed to submitting a version formatted with the official NeurIPS 2025 LaTeX template.

+ I continue to find the idea of modeling the trajectory alignment problem as a maximum weight independent set (MWIS) problem interesting.

Given these points, I intend to retain my positive score.

**Limitations:**

Yes.

**Paper Formatting Concerns:**

The article's formatting style does not adhere to the official NeurIPS 2025 template, which appears inconsistent with ‌NeurIPS 2025 style guidelines

**Quality:**

3

**Strengths And Weaknesses:**

Strengths:

1．	The idea of modeling the trajectory alignment problem as an MWIS problem is interesting and provides a novel perspective on solving the trajectory imitation challenge.

2．	The paper provides a thorough theoretical analysis, proving the NP-completeness of the general problem and presenting efficient algorithms under practical assumptions.

3．	Experiments on grid-world tasks demonstrate the effectiveness of the proposed framework compared to standard imitation learning methods.


Weaknesses:

1．	The experiments are conducted on relatively simple grid-world tasks. More experiments on larger or continuous MDPs are needed to fully evaluate the scalability and robustness of the proposed method.

2．	The experimental results lack error bars or other statistical significance measures, which makes it difficult to assess the consistency and reliability of the reported performance.

3．	The paper does not appear to have been compiled using the official NeurIPS 2025 LaTeX template, which is a requirement for submission.

---

> ### Author Rebuttal · Authors · 2025-07-31
>
> We thank the reviewer for recognizing the value of our work and providing
> valuable feedback. We address the concerns of the reviewer as follows.
>
> $\textbf{Q1.}$ “The experiments are conducted on relatively simple grid-world tasks. More experiments on larger or continuous MDPs are needed to fully evaluate the scalability and robustness of the proposed method."
>
> $\textbf{A1.}$ We appreciate the reviewer’s suggestion to evaluate our proposed TGL framework in more complex and real-world environments to further assess its scalability and robustness. First, we emphasize that the theoretical foundation of TGL is built upon a Markov Decision Process (MDP) with a finite number of states and actions, as well as a finite set of expert trajectories. Therefore, our current framework does not extend to continuous MDPs or settings involving an infinite number of expert trajectories. Regarding larger environments, our experimental objective is not to showcase raw performance in high-dimensional state and action spaces, but to isolate and validate TGL’s capacity to solve the trajectory alignment problem via the novel MWIS-based method. Scaling the grid to larger sizes—such as $8 \times 8$ or $10 \times 10$—increases the number of states by a constant factor but does not introduce qualitatively new types of trajectory conflicts. As a result, the core graph learning structure and learning dynamics of TGL remain consistent.
>
>
> $\textbf{Q2.}$ “The experimental results lack error bars or other statistical significance measures, which makes it difficult to assess the consistency and reliability of the reported performance."
>
> $\textbf{A2.}$ We appreciate the reviewer for pointing this out. Our primary metric, the empirical trajectory match probability, is a binomial proportion estimated over 10,000 episodes per evaluation, and each episode is essentially a Bernoulli trial: the generated trajectory either matches or doesn't match a demonstration trajectory. In the worst case: $p = 0.5$, the standard error is $\sqrt{0.5(1-0.5)/10000} = 0.005$, which gives a 95\% confidence interval of 0.01. Because of this, we originally decided to omit error bars as the extra numbers might add clutter to our paper. Nevertheless, we recognize that explicit intervals and better explanations make our results more reliable. Our method outperforms BC confidently when the trajectory set is large, and edges past BC when the trajectory set is small, which is expected because of the noise present in the environment (all of the methods are likely close to the maximum probability to match the demonstration trajectories). We will include these details in our revised version of the paper. Additionally, we have revised the Table 1 results in our paper using the empirical match rate as an estimate for p in standard error:
>
> **Table 1 – Trajectory-match probability (↑ better) over 10 000 episodes.
> Rows indicate the number of deterministic vs. stochastic demos in 𝒟.**
>
> | Expert-labelled set | TGL-UCB (Ours) | BC | PPO expert |
> |---------------------|---------------|----|------------|
> | 15 det / 5 stoch | **0.794 ± 0.008** | 0.775 ± 0.008 | 0.778 ± 0.008 |
> | 10 det / 10 stoch | **0.801 ± 0.008** | 0.778 ± 0.008 | 0.800 ± 0.008 |
> | 5 det / 15 stoch | **0.801 ± 0.008** | 0.771 ± 0.008 | 0.793 ± 0.008 |
> | 10 det / 5 stoch | **0.814 ± 0.008** | 0.793 ± 0.008 | 0.803 ± 0.008 |
> | 8 det / 7 stoch | **0.810 ± 0.008** | 0.774 ± 0.008 | 0.801 ± 0.008 |
> | 5 det / 10 stoch | 0.779 ± 0.008 | 0.758 ± 0.009 | **0.780 ± 0.008** |
> | 10 det only | **0.778 ± 0.008** | 0.753 ± 0.009 | 0.775 ± 0.009 |
> | 5 det / 5 stoch | **0.740 ± 0.009** | 0.713 ± 0.009 | 0.735 ± 0.009 |
> | 10 stoch only | 0.738 ± 0.009 | 0.714 ± 0.009 | **0.739 ± 0.009** |
> | 5 det only | **0.730 ± 0.009** | 0.703 ± 0.009 | 0.713 ± 0.009 |
> | 5 stoch only | **0.675 ± 0.009** | 0.673 ± 0.009 | 0.661 ± 0.009 |
> | 3 det only | **0.672 ± 0.009** | **0.672 ± 0.009** | 0.670 ± 0.009 |
> | 3 stoch only | **0.657 ± 0.009** | 0.653 ± 0.010 | 0.648 ± 0.010 |
> | 1 det only | **0.634 ± 0.010** | 0.627 ± 0.010 | 0.621 ± 0.010 |
>
>
> $\textbf{Q3.}$ “The paper does not appear to have been compiled using the official NeurIPS 2025 LaTeX template, which is a requirement for submission."
>
> $\textbf{A3.}$ We sincerely apologize for not using the official NeurIPS 2025 LaTeX template in our initial submission. We appreciate the reviewer’s attention to this requirement. We will promptly recompile the paper using the correct template and ensure that all formatting guidelines are strictly followed in the camera-ready version, should the paper be accepted. Thank you for pointing this out.
>
> $\textbf{Q4.}$ “Can the authors conduct experiments on more complex and real-world environments to further validate the scalability and robustness of the proposed TGL framework?"
>
> $\textbf{A4.}$ We kindly refer the reviewer to our response to Question 1 (Answer 1), where we have addressed this point in detail.
>
> $\textbf{Q5.}$ “Can the authors report error bars or perform statistical significance tests on their experimental results to provide a more rigorous evaluation of the performance of TGL compared to baselines?"
>
> $\textbf{A5.}$ We kindly refer the reviewer to our response to Question 2 (Answer 2), where we have addressed this point in detail.
>
> $\textbf{Q6.}$ “Can the authors provide more details on the practical relevance and validity of the structural assumptions (bounded realizability of expert trajectories, tree-structured MDP) made in their theoretical analyses?"
>
> $\textbf{A6.}$ Thank you for your question. We explain the practical relevance and validity of structural assumptions in our theory here.
>
> A clear example of the bounded realizability assumption can be found in the OpenAI Gym's CartPole or MuJoCo locomotion benchmarks. In these settings, expert trajectories are often generated by a pre-trained high-performing policy (e.g., PPO or SAC). Since these policies reliably solve the task, each recorded expert trajectory corresponds to a successful rollout that occurs with consistent, non-negligible probability. Importantly, because the expert dataset is finite and curated—often by sampling from stable policy rollouts—all trajectories in the set are not only reachable but also share a uniform lower bound on their likelihood under the environment dynamics. This matches the bounded realizability assumption: each trajectory has at least some positive probability of being sampled during interaction, ensuring that the learning agent has a realistic chance of encountering similar trajectories during training.
>
> For tree-structured MDPs, we kindly refer the reviewer to the practical motivation provided in lines 264–269. We reiterate the relevance here for clarity. Tree-structured MDPs naturally arise in scenarios where future decisions evolve independently and paths do not reconverge. For instance, in large language models (LLMs), decoding strategies such as beam search or top-k sampling generate diverse continuations from a given prompt, forming a forward tree where each branch corresponds to a distinct output sequence. Similarly, in autonomous driving, planning algorithms simulate future actions—such as turning or accelerating—under physical and safety constraints that prevent the system from revisiting prior states, yielding a branching structure of possible, non-overlapping trajectories.

---

### Note · Authors · 2025-08-14

$\textbf{Author Final Remarks}$

We sincerely thank the ACs, SACs, and reviewers for their thoughtful feedback. Our paper tackles a core challenge in RL without reward design: faithfully preserving long-horizon expert behavior through direct trajectory alignment(DTA).

$\textbf{Our Contributions}$

1. $\textbf{Theoretical.}$ We formalize DTA, prove NP-completeness via an MWIS reduction, and identify two tractable structural cases ($\varepsilon$-realizable trajectory sets and Tree MDPs) with polynomial-time solutions. For unknown dynamics, we propose TGL-UCB with a gap-dependent sublinear regret bound.

2. $\textbf{Methodological.}$ We propose the TGL framework, which builds a trajectory conflict graph, solves MWIS to select trajectories, and derives a policy to maxmimize the likelihood of reproducing them. In the unknown-model case, TGL-UCB couples UCB-based exploration with the conflict-graph structure to balance exploration and exploitation.

3. $\textbf{Empirical.}$ On a $4 \times 4$ Frozen Lake variant, TGL-UCB consistently surpasses BC and often the PPO expert in $\textit{trajectory match probability}$, especially as the demonstration set grows large and diverse.

$\textbf{Improvements made during rebuttal}$

1. Motivation: Clearer comparisons to IRL, PbRL, and IL, clarifying why they fail to ensure full-trajectory fidelity, and illustrating when the DTA objective is preferable.

2. Theory: Refined definitions (conflict in trajectories, $\varepsilon$-realizable, Tree MDP) and provided more practical relevance of our assumptions.

3. Experiment-theory link: Justified trajectory match probability as the natural evaluation metric in DTA setting, and explained gains with noisy and diverse demos.

4. Formatting: Aligned with the NeurIPS 2025 $\LaTeX{}$ template, corrected minor typos, and notation.

5. Limitations \& future work: Expanded discussion on limitations, such as computational cost of MWIS in large environments, and outlined future directions for scaling TGL to continuous MDPs and better benchmarks.

6. Uncertainty: Added Wald interval as error bars to explicitly quantify variability across runs.

These additions strengthen both the theoretical foundations and empirical relevance of our work, directly addressing all major reviewer concerns. TGL offers a principled, structure-aware approach to long-horizon decision-making when the goal is to reproduce expert trajectories rather than to infer or optimize rewards.

---

### Decision · Program_Chairs · 2025-09-17

**Decision:**

Accept (spotlight)

**Comment:**

This paper considers the problem of learning policies to mimic expert trajectories. The core issue with existing behavior cloning like methods is that even if fit perfectly to expert data, the learned policy can result in trajectories which are not supported under the expert distribution. The root cause of this can be traced to the step-by-step nature of behavior-cloning. To avoid this, they aim to do imitation at trajectory level. While imitation of the trajectory distribution itself  has been studied exhaustively (e.g., trajectory distribution matching objectives in convex MDP literature [Table1, 1]), the core contribution of the work is in establishing hardness of the problem and then providing assumptions under which one can hope to solve this problem.

The core idea is to frame the direct trajectory alignment problem as a Maximum-Weight Independent Set (MWIS) problem on a graph, which is known to be NP-complete. In this "conflict graph," trajectories are nodes, and an edge connects two trajectories if they require conflicting actions at the same state. To make the problem feasible, authors assume bounded realizability and tree-structured MDPs. For the setting where environment dynamics are unknown, paper proposes UCB style algorithm to achieve sub-linear regret.

Overall, the reviewers appreciated the novelty of the work and found the reduction to existing NP-complete problem interesting. The provided experiments are extremely basic, so left more for asking. However, given the foundational nature and the theoretical focus of the work, I believe the merits of the paper still prevail.

Some additional recommendations

- Consider moving Appendix A.2 into the main paper, since it makes the problem statement crystal clear for a broader audience.
- One could argue that the problem is due to ill-specified/partially-observed state space, and behavior cloning with history conditioned state (as done in practice often) could address the mentioned challenge. Having some discussion about when and how the proposed method could be relevant in such settings as well would be beneficial.

[1]  Mutti, Mirco, et al. "Challenging common assumptions in convex reinforcement learning." *Advances in Neural Information Processing Systems* 35 (2022): 4489-4502